# noisi: A Python tool for ambient noise cross-correlation modeling and noise source inversion

Laura Ermert[1, 5], Jonas Igel[2, *], Korbinian Sager[3, *], Eléonore Stutzmann[4], Tarje Nissen-Meyer[5], and Andreas Fichtner[2]

[1]Department of Earth and Planetary Sciences, Harvard University, 24 Oxford Street, Cambridge, Massachusetts 02139, USA
[2]Institut für Geophysik, ETH Zürich, 8092 Zürich, Switzerland
[3]Earth, Environmenal and Planetary Sciences, Brown University, Providence, Rhode Island 02912, USA
[4]Université de Paris, Institut de Physique du Globe de Paris, CNRS, F-75005 Paris, France
[5]Department of Earth Sciences, University of Oxford, Oxford OX1 3AN, UK
[*]These authors contributed equally to this work.

**Correspondence:** Laura Ermert (lermert@fas.harvard.edu)

**Abstract.** We introduce open-source tool `noisi` for the forward and inverse modeling of ambient seismic cross-correlations with spatially varying source spectra. It utilizes pre-computed databases of Green's functions to represent seismic wave propagation between ambient seismic sources and seismic receivers, which can be obtained from existing repositories or imported from the output of wave propagation solvers. The tool was built with the aim of studying ambient seismic sources while accounting for realistic wave propagation effects. Furthermore, it may be used to guide the interpretation of ambient seismic auto- and cross-correlations, which have become pre-eminent seismological observables, in light of non-uniform ambient seismic sources. Written in the Python language, it is both accessible for usage and further development, as well as efficient enough to conduct ambient seismic source inversions for realistic scenarios. Here, we introduce the concept and implementation of the tool, compare its model output to cross-correlations computed with SPECFEM3D_globe, and demonstrate its capabilities on selected use cases: A comparison of observed cross-correlations of the Earth's hum to a forward model based on hum sources from oceanographic models, and a synthetic noise source inversion using full waveforms and signal energy asymmetry.

## 1 Introduction

### 1.1 Motivation

Cross-correlations of ambient seismic noise form the basis of many applications in seismology, from site effects studies (e.g., Aki, 1957; Roten et al., 2006; Bard et al., 2010; Denolle et al., 2013; Bowden et al., 2015) to ambient noise tomography (e.g., Shapiro et al., 2005; Yang et al., 2007; Nishida et al., 2009; Haned et al., 2016; de Ridder et al., 2014; Fang et al., 2015; Singer et al., 2017) and coda wave interferometry (e.g., Sens-Schönfelder and Wegler, 2006; Brenguier et al., 2008; Obermann et al.,

2013; Sánchez-Pastor et al., 2019). Auto-correlations of the ambient noise are also increasingly used to study seismic interfaces
as suggested by Claerbout (1968) (e.g., Taylor et al., 2016; Saygin et al., 2017; Romero and Schimmel, 2018) and to monitor
subsurface properties (Viens et al., 2018; Clements and Denolle, 2018).

Importantly, most ambient noise studies are based on the assumption that noise cross-correlations converge to inter-station
Green's functions (Weaver and Lobkis, 2001; Shapiro and Campillo, 2004; Wapenaar, 2004), which is in general not fulfilled
(e.g. Halliday and Curtis, 2008; Kimman and Trampert, 2010; Stehly et al., 2008; Sadeghisorkhani et al., 2017). Numerical
models of noise auto- and cross-correlations allow us to probe this assumption and eventually circumvent it (Halliday and
Curtis, 2008; Fan and Snieder, 2009; Cupillard and Capdeville, 2010; Kimman and Trampert, 2010; Fichtner, 2014; Stehly
and Boué, 2017; Delaney et al., 2017). While the number of applications based on the Green's function assumption is large
and rapidly increasing (Nakata et al., 2019), only a modest number of studies have presented models of ambient noise cross-
correlations themselves, i.e. numerical evaluations of cross-correlations due to distributed noise sources, rather than models
of Green's functions (e.g., Nishida and Fukao, 2007; Tromp et al., 2010; Hanasoge, 2013a; Basini et al., 2013; Ermert et al.,
2017; Sager et al., 2018b; Datta et al., 2019; Xu et al., 2018, 2019; Sager et al., 2020).

Several state-of-the-art open-source tools for ambient noise data processing are freely available, e.g., Whisper (https://code-
whisper.isterre.fr/, July 7, 2020), MSnoise (Lecocq et al., 2014), FastPCC (Ventosa et al., 2019), yam (https://github.com
/trichter/yam, July 7, 2020) and NoisePy (https://github.com/chengxinjiang/Noise_python, July 7, 2020). However, the same
cannot be said about cross-correlation modeling tools, which have mostly been developed ad hoc by different research groups
(Hanasoge, 2013a; Fichtner, 2014; Sager et al., 2020; Xu et al., 2019). An exception is the openly available implementation
of noise cross-correlations and sensitivity kernels in SPECFEM3D (Tromp et al., 2010); however, in its current form it is not
tailored to the exploration of different noise source models and their impact on cross-correlation. Moreover, it requires high
performance computing (HPC) resources for many applications.

Therefore, we present a tool named `noisi` for modeling ambient noise cross-correlations while honoring the physics of wave
propagation, and for determining source sensitivity kernels which can be used for rapid, cross-correlation-based ambient noise
source inversion. The tool is implemented in Python, parallelized using mpi4py (Dalcín et al., 2005) and provided on github,
alongside a tutorial and an exemplary ambient noise source inversion setup. In the following paper, we describe the ideas
behind `noisi` and its implementation, compare its output to cross-correlations modeled with SPECFEM3D_GLOBE, and
illustrate its current capabilities with selected use cases.

## 1.2  Using waveform databases for rapid, realistic cross-correlation models

One of the main challenges in modeling ambient noise cross-correlations is the adequate representation of seismic wave prop-
agation from the noise sources, which are in general globally distributed (Stehly et al., 2006; Nishida and Takagi, 2016;
Retailleau et al., 2018), to seismic receivers. The noise cross-correlation implementations of Tromp et al. (2010) and Sager
et al. (2018a) honor the physics of wave propagation to the greatest possible extent, but require substantial HPC resources for
inversion (Sager et al., 2020). The `noisi` tool uses databases of pre-calculated seismic wavefields instead to compute cross-
correlations and sensitivity kernels. It therefore presents an alternative for cross-correlation modeling and noise source inversion

for cases where updates to the structure model (i.e., seismic velocities, density, and attenuation) are not required.Owing to the reuse of Green's functions, computation is quick and inexpensive. However, storage resources, typically on the order of 1 GB per station, are needed to hold the Green's function database.

Databases of pre-calculated Green's functions have recently been applied to a variety of seismological problems, such as source inversion of earthquakes (Dahm et al., 2018; Fichtner and Simutė, 2018), landslides (Gualtieri and Ekström, 2018) and ambient noise (Ermert et al., 2017; Datta et al., 2019). Although the generation of such databases themselves often requires HPC resources, they can be shared to provide access to the results of costly wave propagation simulations to users without access to those resources. This is achieved, for example, by the IRIS Synthetics engine (Syngine) repository (IRIS, 2015; Krischer et al., 2017) and by tools for the extraction and management of Green's function databases (van Driel et al., 2015a; Heimann et al., 2019). The `noisi` tool enables the use of Syngine databases for modelling noise cross-correlations. However, it is not limited to these; rather, pre-calculated Green's function databases from any numerical wave propagation solver, which may include 3-D Earth structure, topography, etc. can be used with `noisi` after appropriate formatting.

## 1.3 Possible applications

Various examples fall within the range of possible applications of `noisi`. For example, it can be used to probe the quality of Green's functions retrieved from noise cross-correlations in a variety of different source scenarios, such as previously studied in simplified models, e.g. by Halliday and Curtis (2008); Kimman and Trampert (2010) and Fichtner (2014). Furthermore, the influence of noise sources on the reliability of scattering and attenuation measurements can be studied, again previously explored by Fan and Snieder (2009); Stehly and Boué (2017) and Nie et al. (2019). In addition, ground motion auto-correlations, i.e. power spectral densities of seismic noise, can be modeled for arbitrary noise source distributions. Finally, it can be utilized for noise source inversion when no updates to the Earth structure model are required, similar to the pioneering study by Nishida and Fukao (2007), who inverted observed cross-correlations for source distribution of the Earth's hum, and as performed by Ermert et al. (2017); Xu et al. (2018, 2019) and Datta et al. (2019).

## 2 Cross-correlation modeling

Ambient seismic noise can be considered as the superposition of elastic waves that have propagated from various traction sources $\boldsymbol{N}(\boldsymbol{\xi}, \omega)$ at Earth's surface $\partial\oplus$. The amplitude of the sources depends on location $\boldsymbol{\xi}$ and frequency $\omega$; their complex phase is treated as random variable. One component of ground motion $u_i$ observed at a seismic receiver at location $\boldsymbol{x}$ can be modeled as the convolution of the noise source time series with the impulse response of the Earth or Green's function $G$. In frequency domain, this relation is expressed as

$$u_i(\boldsymbol{x}, \omega) = \int_{\partial\oplus} G_{in}(\boldsymbol{x}, \boldsymbol{\xi}, \omega) N_n(\boldsymbol{\xi}, \omega) \, \mathrm{d}\boldsymbol{\xi}, \tag{1}$$

(Aki and Richards, 2002), where summation over repeated indices is implied. The correlation of two such signals, averaged over an observation period, can be expressed by multiplication in the frequency domain, i.e.


$$\mathcal{C}_{ij}(\boldsymbol{x}_1, \boldsymbol{x}_2, \omega) = \langle u_i^*(\boldsymbol{x}_1, \omega) u_j(\boldsymbol{x}_2, \omega) \rangle$$

$$= \left\langle \iint_{\delta \oplus} G_{in}^*(\boldsymbol{x}_1, \boldsymbol{\xi}_1, \omega) N_n^*(\boldsymbol{\xi}_1, \omega) G_{jm}(\boldsymbol{x}_2, \boldsymbol{\xi}_2, \omega) N_m(\boldsymbol{\xi}_2, \omega) \, \mathrm{d}\boldsymbol{\xi}_1 \mathrm{d}\boldsymbol{\xi}_2 \right\rangle, \tag{2}$$

where $\langle \rangle$ denotes time-averaging and the correlation is written as convolution with the time-reversed, or complex conjugate signal as indicated by the $^*$. We adopt an integral description here, as we assume that the noise sources $N(\boldsymbol{\xi}_1, \omega)$ and $N(\boldsymbol{\xi}_2, \omega)$ are generally extended and vary continuously over more or less extended source areas. Equation 2 only assumes that seismic

signals at the receivers are predominantly seismic waves, and that further observational noise, such as instrument tilt, has been removed or is expected to be incoherent in the cross-correlation.

Noise cross-correlation modeling hence has to address how to parametrize the noise sources $N_m(\boldsymbol{\xi}, \omega)$ and how to model the propagation of their signals to receivers ($G_{jm}(\boldsymbol{x}, \boldsymbol{\xi}, \omega)$). To deal with sources of unknown, stochastic phase, it is commonly assumed that they are spatially uncorrelated when averaged over a sufficiently long observation span, or that their correlation

length is far below observational resolution (e.g. Snieder, 2004; Nishida and Fukao, 2007; Tromp et al., 2010; Stutzmann et al., 2012; Hanasoge, 2013b; Farra et al., 2016; Xu et al., 2018; Datta et al., 2019). Upon this assumption, the noise sources can be described by their location-dependent power spectral density (PSD):

$$\langle N_n^*(\boldsymbol{\xi}_1, \omega) N_m(\boldsymbol{\xi}_2, \omega) \rangle = S_{nm}(\boldsymbol{\xi}_1, \omega) \delta(\boldsymbol{\xi}_1 - \boldsymbol{\xi}_2) \tag{3}$$

removing the requirement to model their phase. Assuming that the change of Green's functions in between observation windows

is negligible, Equation 2 can be rearranged so that the source PSD can be substituted and the cross-correlation becomes:

$$\mathcal{C}_{ij}(\boldsymbol{x}_1, \boldsymbol{x}_2, \omega) = \int_{\delta \oplus} G_{in}^*(\boldsymbol{x}_1, \boldsymbol{\xi}, \omega) G_{jm}(\boldsymbol{x}_2, \boldsymbol{\xi}, \omega) S_{nm}(\boldsymbol{\xi}, \omega) \, d\boldsymbol{\xi}, \tag{4}$$

which greatly simplifies the model. The sources $N_n, N_m$ are traction sources as mentioned above, so that $S_{nm}$ can be regarded as a power spectral density of pressure at the Earth's surface, with units of $\mathrm{Pa}^2\mathrm{s}$. Importantly, ambient seismic source amplitudes usually vary with observation period. For example, oceanic sources show both short-term and seasonal variations

(Ardhuin et al., 2011; Stutzmann et al., 2012). Therefore, the cross-correlations $\mathcal{C}$ generally depend on the time and duration of observation. Often, such time dependence due to source variability is regarded as a nuisance effect in ambient noise studies and stacks are formed to mitigate this effect (e.g. Stehly et al., 2009). However, we illustrate and discuss below how using modeling tools such as `noisi` enables us to incorporate source information for an extended interpretation of what signals cross-correlations may contain.

For the evaluation of equation 4, source-receiver reciprocity (Aki and Richards, 2002) is invoked

$$G_{jm}(\boldsymbol{x}_2, \boldsymbol{\xi}, \omega) = G_{mj}(\boldsymbol{\xi}, \boldsymbol{x}_2, \omega) \tag{5}$$

so that a point force source can be placed at the location of one seismic receiver, and the Green's functions to any source at the Earth's surface is recorded, which is far more practicable than simulating waves from a large number of possible seismic noise source locations to the receiver.

If an Earth model is assumed a priori, e.g. the Preliminary Reference Earth Model (Dziewoński and Anderson, 1981) or another model resulting from seismic tomography, the obtained Green's functions $G_{in}(\boldsymbol{x}_1, \boldsymbol{\xi}, \omega)$ and $G_{jm}(\boldsymbol{x}_2, \boldsymbol{\xi}, \omega)$ are fixed throughout the simulation or inversion, and equation 4 can be evaluated multiple times while requiring only one potentially costly wave propagation simulation per receiver, or none if prepared databases such as the ones from Syngine are used. This strategy is implemented in the `noisi` tool.

Similar to the derivation of the forward model, the misfit gradient with respect to noise source parameters, which is needed for noise source inversion, can be obtained. For one receiver pair and components $i, j$, the misfit sensitivity kernel is given by

$$K_{nm}(\boldsymbol{x}_1, \boldsymbol{x}_2, \boldsymbol{\xi}) = \int\limits_{\omega_0}^{\omega_1} G_{in}^*(\boldsymbol{x}_1, \boldsymbol{\xi}, \omega) G_{jm}(\boldsymbol{x}_2, \boldsymbol{\xi}, \omega) f_{ij}(\boldsymbol{x}_1, \boldsymbol{x}_2, \omega) \, d\omega, \tag{6}$$

where $f(\boldsymbol{x}_1, \boldsymbol{x}_2, \omega)$ depends on the chosen measurement function used to compare modeled and observed noise cross-correlations and the last two indices of the kernel, $nm$, refer to the source (cross-)components. The misfit gradient can then be compiled

as a sum of sensitivity kernels. For details on kernels and inversion, the interested reader is referred to Tromp et al. (2010); Hanasoge (2013a); Fichtner (2014); Ermert et al. (2017); Sager et al. (2018b) and Xu et al. (2019).

The `noisi` tool computes both forward model and sensitivity kernels. It has been constructed to fulfill both tasks in a simple, flexible, and computationally inexpensive way, and addresses them as follows. Noise sources are treated as spatially varying power spectral densities according to equation 3. Wave propagation Green's functions $G_{in}$ are read from a database in hdf5-

format (Folk et al., 2011). The tool includes setup routines for i) analytic surface wave Green's functions, ii) Green's functions for spherically symmetric Earth models provided by `instaseis` (van Driel et al., 2015b) with wavefield databases hosted by the Syngine repository (Krischer et al., 2017), and iii) Green's functions for laterally varying Earth models obtained from AxiSEM3D (Leng et al., 2016, 2019). The AxiSEM3D solver can account for a number of pre-defined mantle models, topography and crustal model crust1.0 (Bassin et al., 2000). In addition to these three options, custom waveform databases can be

constructed by formatting results of wave propagation modelling as an hdf5 database usable by `noisi`.

## 3   Implementation

The core tasks of the tool are to evaluate equations 4 and 6. This is done by approximating the integrals by a weighted sum:

$$\mathcal{C}_{ij}(\boldsymbol{x}_1, \boldsymbol{x}_2, \omega) = \int\limits_{\delta\oplus} G_{in}^*(\boldsymbol{x}_1, \boldsymbol{\xi}, \omega) G_{jm}(\boldsymbol{x}_2, \boldsymbol{\xi}, \omega) S_{nm}(\boldsymbol{\xi}, \omega) \, d\boldsymbol{\xi},$$

$$\approx \sum_{s=1}^{n_s} [G_{in}^*(\boldsymbol{x}_1, \boldsymbol{\xi}_s, \omega) G_{jm}(\boldsymbol{x}_2, \boldsymbol{\xi}_s, \omega) S_{nm}(\boldsymbol{\xi}_s, \omega)] \, \Delta\boldsymbol{\xi}_s, \tag{7}$$

and accordingly for equation 6. Additionally, the model contains setup scripts creating the source model, wavefield database, handling directory structure and measurements. The module is command-line based for convenient calling and scripting. Its

computational tasks mostly rely on numpy (Oliphant, 2006). PyYaml (Simonov, 2014) is used to handle readable and commented configuration files, scipy (Millman and Aivazis, 2011) is used for signal processing tasks, obspy (Beyreuther et al., 2010) for signal processing, geodetic functions, and access to seismic data formats, h5py (Collette, 2013) for the handling of the hdf5 format, mpi4py (Dalcín et al., 2005) for parallelization using MPI and cartopy for basic plotting. The installation of `instaseis` (van Driel et al., 2015b) is optional, and allows users to obtain Green's functions from reciprocal or merged `instaseis` databases which can, for example, be downloaded from the Syngine repository (Krischer et al., 2017). Below we briefly describe the implementation in more detail following a possible sequence of work to create a cross-correlation model and noise source inversion.

## 3.1 Definition of source model grid

The discretized noise source grid that will be used throughout modeling and inversion is pre-defined and fixes the locations of possible noise sources. For each evaluation of equations 4 and 6, Green's functions $G_{in}$ and source spectra $S_{nm}$ at locations $\boldsymbol{\xi}_s$ are matched by index. This reduces computational effort during modeling and inversion. Grid setup aims to collect locations of approximately equal surface area around each point on the surface of the WGS84 ellipsoid. This is achieved by selecting points at equal distance (in meters) in latitudinal and longitudinal direction. The parameters to be specified by the user are grid step, as well as minimum and maximum coordinates. An example for a regional grid is shown in Figure 1, panel e).

Since the rectangle rule is used for spatial integration (Eq. 7), a finer grid reduces integration error. For the comparison to SPECFEM3D_globe (shown below), the spacing is chosen as one half of the shortest expected seismic wave length, while for the synthetic inversion in section 5.2, it is set to one quarter of the shortest wave length. Either rule of thumb produces satisfactory results, although small improvements are obtained using the finer spacing (see supplement). To exclude that integration errors severely affect the modeled cross-correlations, testing the convergence of the results with decreasing grid step is recommended, in particular when body waves in the cross-correlations are considered. Improvements of spatial integration are the subject of current developments (e.g. Igel, 2019).

The grid only defines source longitude and latitude, but does not specify elevation. The influence of an eventual topography of the underlying wave propagation model on the surface area of each grid cell is neglected. However, topography itself can be taken into account: The Green's functions $G_{in}$ describe propagation from and to the surface of the underlying numerical wave propagation model. Therefore, topography or bathymetry are determined by their value in the respective geographic location of the wave propagation model.

## 3.2 Source model parametrization

Instead of parametrizing the sources as fully sampled spectra at each grid point, their spectra are represented by a small number of Gaussian functions of frequency in each grid location, which reduces the dimensionality of the model and inverse problem, and ensures that the source PSD in each location is smooth. This is illustrated in Figure 1, which shows an example of a basic source model that may be subsequently updated by noise source inversion. The model contains sources which are homogeneously distributed throughout the ocean (Panel a) as well as a localized source (Panel c); note that their maximum amplitudes

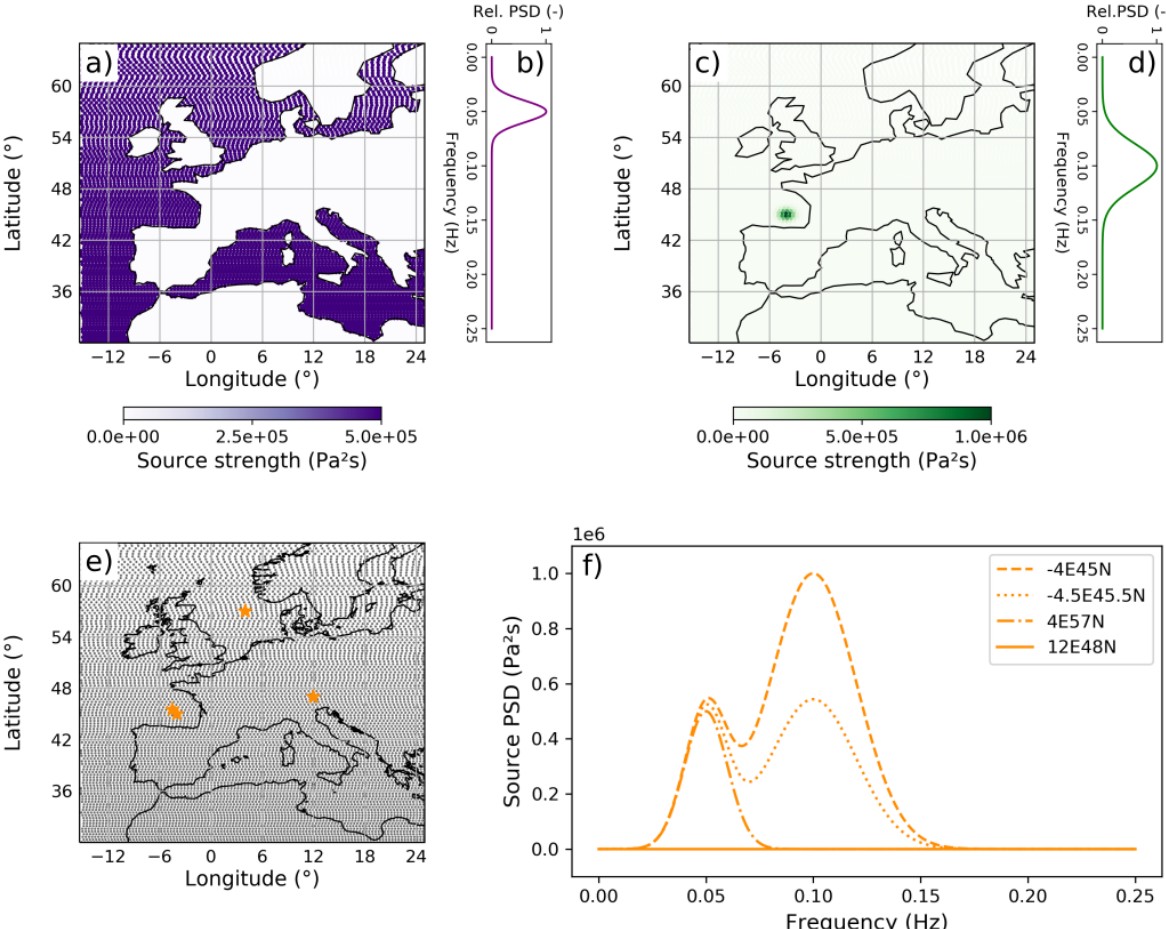

**Figure 1.** Illustration of noise source model parametrization. The upper panels show spatial source distributions (a, c) with different spectra (b, d). Note the difference in maximum amplitudes. (Similar figures can be reproduced by following the Jupyter notebook tutorial for noisi). Panel e shows the grid on which source spectra are defined; power spectral densities, corresponding to the term $S_{nm}$ of equation 4, for the grid locations $\boldsymbol{\xi}$ marked by yellow stars, are shown in panel f. These are the superposition of spectra b and d, with spatially varying amplitudes specified by distributions a and c, respectively.

differ. Each of these distributions are associated with a different amplitude spectrum (Panels b, d). Thus, in any location of the source grid, the effective source spectrum is a superposition of both spectra weighted by their respective distribution. This is shown in panel f, with spectra at locations marked by yellow stars on the map in panel e: On the continent (12E48N), the spectrum is flatly zero, whereas in the North Sea (4E57N) it shows a single peak associated with the source distribution and spectrum in panels a, b. In the Bay of Biscay, the localized strong source of panels c, d, which varies at shorter distance from

-4E45N to -4.5E45.5N, is also visible.

Any number of such distributions can be superimposed to create a source model. Gaussian PSDs and their spatial weights at each grid point are stored in hdf5 format as detailed in the appendix. Examples for all input files are also provided in the github repository. The parameters for setup are geographic distributions, (geographically homogeneous, ocean, and Gaussian 'blob'), as well as the central frequency and variance of the Gaussian spectra. Custom source models can be created by modifying the

underlying hdf5 file (an example is shown in section 5.1).

### 3.3 Wavefield databases

Green's functions are stored in one hdf5-file per seismic receiver component. The format is specified in the appendix. For the preparation of this database, routines are provided that take a seismic station list, the format of which is also specified in the appendix, as input. One may set up a database for analytic far-field surface wave Green's functions for 2-D homogeneous media

(following Fan and Snieder, 2009); obtaining Green's functions for PREM or other reference models additionally requires an `instaseis` database (e.g. downloaded from Syngine). If a surface wavefield output from AxiSEM3D is provided, Green's functions can be extracted from this surface wavefield, allowing to include 3-D lateral Earth structure. If run on multiple processors, the task of preparing the Green's function database is performed in embarrassingly parallel mode, where each receiver component is prepared on one core.

Custom wavefields can be built by converting the format of previously computed surface wavefields. Similarly to the example of converting from AxiSEM3D output, output from any other wave propagation solver may be interpolated at the grid locations and stored in the hdf5 format as detailed in the appendix for use with the `noisi` tool. Crucially, the hdf5 format (Folk et al., 2011) allows convenient access to single Green's functions. These may be stored either as time series or complex spectra; details on this choice are explained below.

### 3.4 Evaluation of cross-correlations

The tool evaluates correlations for all possible combinations of stations specified in the station list (see appendix) and the selected component, optionally including auto-correlations. If run on multiple processors, tasks are again distributed according to a simple embarrassingly parallel scheme.

While the convolutions of equations 4, 6 are performed in frequency domain for speed, storage of the Green's functions may

be more convenient in time or frequency domain depending on the application; procedures for either domain are implemented. When storage is in the frequency domain, no Fast Fourier transform (FFT) of the Green's functions is needed during calculation, which eases the computation. As the Green's functions are real functions of time, their spectra are Hermitian, so that storing

their non-negative-frequency part suffices to describe them fully. However, the Green's functions have to be zero-padded prior to Fast Fourier Transform in order to preclude circular convolution and to increase frequency resolution. When the Green's functions are stored in time domain, this zero-padding is done on the fly during computation, before FFT is performed. Thus, the number of samples decreases compared to frequency domain storage, resulting in reduced storage and I/O effort despite increased computational effort of performing FFT.

The resulting cross-correlations are saved in SAC format, with essential metadata contained in the SAC header.

### 3.5 Measurements and evaluation of sensitivity kernels

To run noise source inversion, observed auto- and/or cross-correlations must be provided as SAC files, with their headers containing a fixed set of metadata as specified in the appendix (usage similar to IRIS DMC, 2015). Measurements can then be performed on the data and the modeled cross-correlations, yielding a misfit between the current model and the observations. Implemented measurements include windowed and full waveforms, mean squared amplitudes, and the logarithmic signal energy ratio between causal and a-causal correlation branch. For details on these measurements, see Sager et al. (2018a). Running the measurement will additionally determine the term $f(\omega)$ of equation 6, which is frequently referred to as adjoint source. This term corresponds to the derivative of the measurement with respect to the synthetic cross-correlation trace. Sensitivity kernel computation is run analogous to the forward model, i.e. reading in Green's functions for each source location identified by index. Kernels are saved as $\ell$ by m by n-dimensional arrays, where $\ell$ is the number of Gaussian PSD spectra, m the number of applied bandpass filters, and n the number of source locations.

## 4 Comparison to SPECFEM3D_GLOBE

To the best of our knowledge, the only currently available open-source model of noise cross-correlations and their sensitivity kernels was provided by Tromp et al. (2010). Thus, we use their implementation to validate and cross-check the output of forward modeling with `noisi`. The implementation by Tromp et al. (2010) follows a different strategy: It models the cross-correlation wavefield by inserting the inverse Fourier transform of the term $G_{jm}(\boldsymbol{x}_2,\boldsymbol{\xi},\omega)S_{nm}(\boldsymbol{\xi},\omega)$ of equation 4 as source term of the wave propagation simulation, yielding $C_{ij}(\boldsymbol{x},\boldsymbol{x_2},\tau)$ in any location of the model domain.

To compare entirely independently computed ambient noise cross-correlations, we use AxiSEM3D (Leng et al., 2016) to create the pre-computed wavefields, on the basis of which we then compute cross-correlations with `noisi`. These are compared to cross-correlations modelled with SPECFEM3D_GLOBE (Komatitsch and Tromp, 2002b, a) as described by Tromp et al. (2010) and in the SPECFEM3D_GLOBE user manual. To exclude relevant deviations between the models stemming from differences between the spectral element solvers, we omit laterally varying crustal models, since their implementation differs between SPECFEM3D_GLOBE and AxiSEM3D, and consider the elastic case without attenuation. We perform the comparison of cross-correlations at periods of up to 15 s for the spherically symmetric PREM (Dziewoński and Anderson, 1981) and for the laterally varying S40RTS (Ritsema et al., 2011) model. Using PREM, the effects of ocean, ellipticity, topography, rotation and gravity were neglected, while they were included with S40RTS. The ocean was modelled as an effective load in both

solvers, and gravity by the Cowling approximation (Komatitsch and Tromp, 2002a). Supplementary figure S1 illustrates the locations of the stations in the modeling domain, which extends to 20 by 20°, as well as the shear wave velocity perturbation of S40RTS with respect to PREM in this region at 20 km depth. The numerical domain for the solution in SPECFEM3D_GLOBE is set to 40 by 40° with absorbing boundaries; the larger domain is chosen to exclude spurious boundary reflections of surface waves from the lag window of interest. However, noise sources are restricted to act in the same domain as for the other case. In

AxiSEM3D, a method that couples a spectral-element discretisation with a pseudospectral expansion along the azimuth (Leng et al., 2019), we simulate the full desired 3D resolution inside the domain of interest. Rather than using absorbing boundaries as in the simulation with SPECFEM3D_GLOBE, we avoid spurious reflections in AxiSEM3D by using a global computational domain. The azimuthal Fourier expansion is tapered to a minimum of two Fourier coefficients outside of our domain of interest, which strongly reduces the additional compute time accrued due to the global simulation.

As source distribution for this example, we chose a homogeneous distribution of noise with Gaussian spectrum peaking at 20 s period. Figure 2 shows the comparison of cross-correlation waveforms obtained from SPECFEM3D_GLOBE and the combination of AxiSEM3D and `noisi`, interpolated to equal sampling rate and filtered consistently by a second-order Chebysheff lowpass filter. Each waveform is normalized to unity for better visibility; a comparison showing the relative amplitudes can be found in the supplement. The traces are arranged by increasing inter-station distance (not to scale). We observe an excellent

fit of the cross-correlation waveforms. Note that the strong asymmetry of several cross-correlations is an effect of the sources being confined to a bounded domain, an effect which is reproduced consistently by both algorithms. This figure also illustrates the effect of different models on the cross-correlations. The correlations for S40RTS show a delay in the arrival of the dominant surface wave groups that increases with higher frequency, which is partially an effect of using different crustal layers (an averaged single crustal layer was used with PREM), as well as the negative velocity perturbations of S40RTS from PREM in

this region, which are illustrated in Figure S1 in the supplement.

Upon close inspection, deviations of the correlations modeled `noisi` from the SPECFEM3D_globe output are visible. The bottom panels of Figure 2 show these, increased by a factor of 10. We suggest that these result mostly from the approximation of the spatial integral that we adopt in Eq. 7. We corroborate this by varying the spatial sampling (see supplement).

## 5   Example applications

### 5.1   Auto- and cross-correlation forward modeling

Forward modelling of ambient noise auto- and cross-correlations has been employed in a number of studies, for example, to investigate noise sources (e.g. Stutzmann et al., 2012; Gualtieri et al., 2013; Juretzek and Hadziioannou, 2017) or to evaluate the assumption of Green's function retrieval (e.g. Stehly and Boué, 2017). The `noisi` tool implements forward modeling for arbitrary distributions of noise sources with Gaussian spectra. To exemplify this, we model correlations of the Earth's hum at

a selection of receiver locations, based on the model of seismic hum as described by Ardhuin et al. (2015) and implemented in Deen et al. (2018), extended to global hum sources. We use a temporal subset of this space-, time-, and frequency-dependent hum source model, namely a selection of Southern hemisphere winter months (July-September), averaged over the frequency

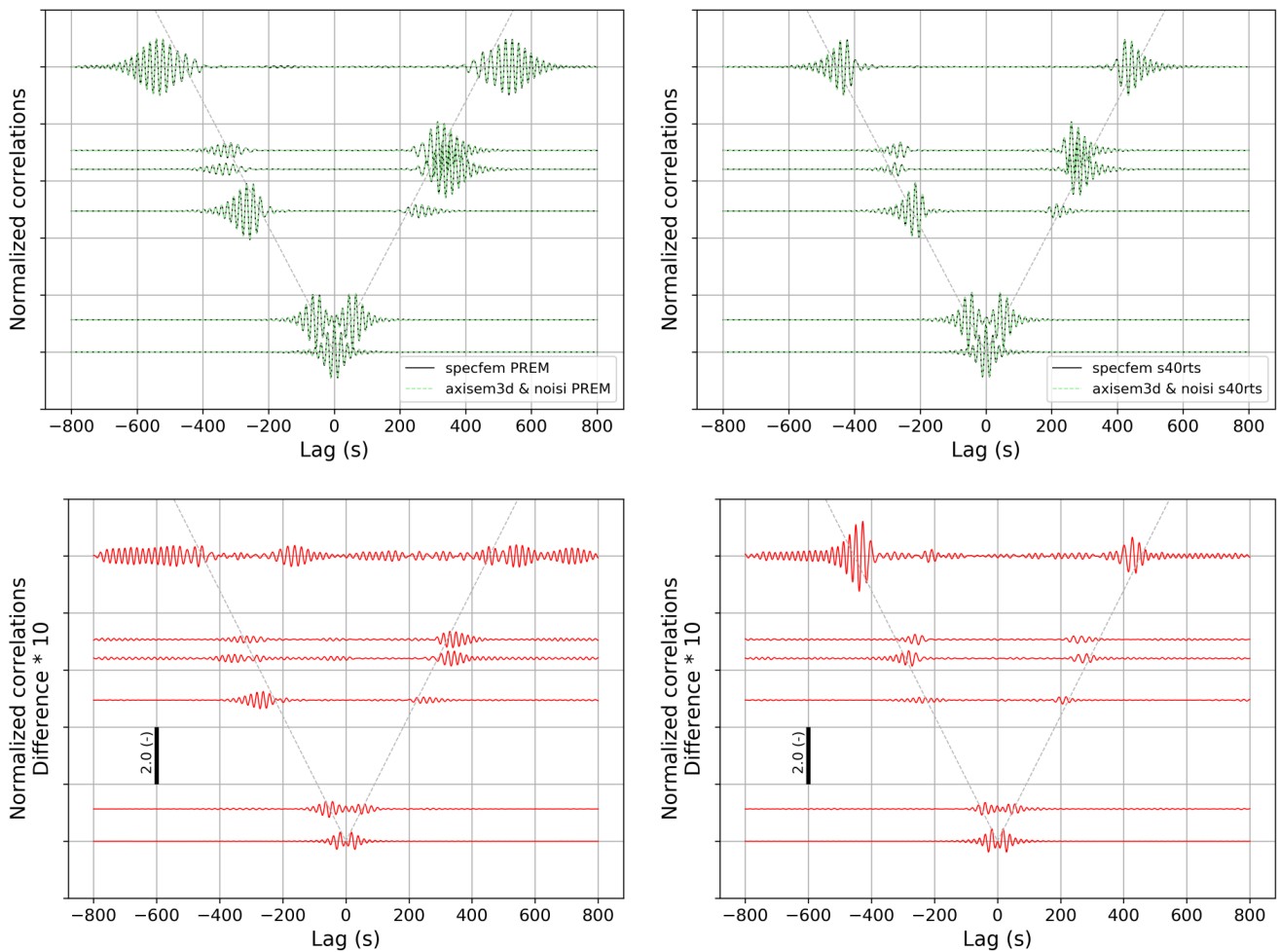

**Figure 2.** Comparison between two implementations of simulation ambient noise cross-correlations with PREM (top left panel) and S40RTS (top right panel). Both panels shows correlations modelled with SPECFEM3D_GLOBE as well as with noisi, where the latter uses Green's functions modelled with AxiSEM3D. A moveout of $3\frac{km}{s}$ is shown by the dashed gray line (note that the y-axis is only approximately to scale for better visibility; however, the cross-correlations for both models are arranged at the same distances). The correlations are lowpass-filtered by type II Chebyshev filter with stopband frequency 0.07 Hz, normalized to unity, and arranged by interstation distance. An overview of the modeling domain, stations, and mantle shear velocity model is shown in the supplement. The bottom panels show the absolute difference between the traces in the top panels, enhanced by a factor of 10. The vertical bars denote the scale.

band 0.0035 - 0.007 Hz. This hum model is interpolated on a dense global source grid for the `noisi` tool. To illustrate the use of Green's functions describing different Earth structures and obtained with different wave propagation solvers, the correlations

are constructed with two different Green's function databases. Synthetics from anisotropic PREM with uniform crustal layer are contrasted with synthetics from S40RTS with attenuation, laterally varying crust2.0, and ocean load. These have been computed with AxiSEM3D (in spherically symmetric mode) and SPECFEM3D_GLOBE, respectively.

We illustrate a selection of the resulting correlations (selected to represent the variety of inter-station paths and distances) in Figure 3. The map shows the averaged source model and station locations for the synthetic correlations. Additional panels show

synthetic correlation traces for two Earth models (orange: PREM, blue: S40RTS+crust2.0). At the long periods considered here, the waveforms of both models are similar, although subtle differences occur both in phase and amplitude. Several cross-correlations show arrivals before the first-arriving Rayleigh wave (the arrival of a surface wave travelling at $3.7\frac{\text{km}}{\text{s}}$ is marked by a red and green dashed line on the a-causal and causal branch, respectively). This occurs, for example, between stations CAN and SSB and stations INU and SSB. These phases, with amplitudes far higher than those expected of fast-travelling

body waves, are due to the source distribution in this synthetic example; similarly early-arriving phases have been previously observed. While sometimes referred to as spurious arrivals, they are physical and can even be utilized for source localization (Retailleau et al., 2017). Generally, the stationary phase of surface waves in the cross-correlation with respect to noise source distribution ensures the retrieval of fundamental mode surface waves from noise cross-correlations (Snieder, 2004; Tsai, 2009). However, the presence of strong or persistent, localized sources off the great circle path which connects the two receivers can

give rise to arrivals before the expected surface waves (e.g., Shapiro et al., 2006; Zheng et al., 2011), appearing approximately at the differential travel time from the source location to the two receivers. Modeling cross-correlations such as the ones in Figure 3 opens up possibilities to study them in more detail, which will possibly enable us to utilize valuable information which might otherwise be discarded as incoherent "noise".

In a further step, we compare the model to observed cross-correlations. Since stacking duration was only 3 months for the

noise source model (July-September 2013), only few of the modeled station pairs yield cross-correlation with acceptable signal-to-noise ratio. These are pairs of stations which are i) exceptionally quiet in the hum band, according to probabilistic power spectral densities for the respective time period, and ii) at moderate or near-antipodal distance to enhance station-to-station surface wave amplitude. These criteria are fulfilled by CAN, SSB and TAM. We show a comparison of their observational cross-correlations with the modeled ones in Figure 4. Cross-correlations were computed in windows of 12 hours with 50 %

overlap after removal of any earthquake with $M_w > 5.6$ as classical, geometrically normalized cross-correlations according to equation 1 of Schimmel et al. (2011), and stacked. All waveforms in Figure 4 are normalized by maximum amplitude. For better visibility, windows around the R1-wave are enlarged. Upon measuring the L2-waveform difference between observed and modeled cross-correlation within these windows, a slightly better overall fit is obtained by using a 3-D Earth model (this holds both for the three correlations selected here, and the collection of all modeled correlations).

The observed cross-correlations are noisy due to the relatively short stack (up to 92 days depending on data availability); cross-correlations in this frequency band are expected to predominantly show direct, fundamental mode surface waves between two stations only after a stacking duration of one year and more (Haned et al., 2016). The observed traces here may contain

incidental, non-coherent apparent correlation, i.e. 'noise of the noise', such as the strong arrival at $\Delta t \cong 500$s on the a-causal zoom of G.CAN–G.SSB. More elaborate stacking schemes (e.g., Schimmel et al., 2011; Ventosa et al., 2019), which are out of the scope of this work, can reduce such effects. It is important to note, however, that similar-looking phases may also be produced by the inhomogeneous source distribution such as the modeled arrival at $\Delta t \cong -300$s on the zoomed causal panel of G.CAN–G.TAM. Modeling can enable us to distinguish and interpret such phases.

## 5.2 Ambient noise source inversion

Sensitivity kernels computed with `noisi` can be used to run gradient-based inversion for the distribution of ambient seismic sources from a data set of observed ambient noise cross-correlations. To demonstrate the effectiveness of this approach, we conduct two synthetic inversions using two different functions to measure the misfit between observations and model. The sensitivity kernel of any misfit function can be expressed as

$$K_{zz}(\boldsymbol{x}_1, \boldsymbol{x}_2, \boldsymbol{\xi}) = \int\limits_{\omega=0}^{\omega_{Nyq}} G_{zz}^*(\boldsymbol{x}_1, \boldsymbol{\xi}, \omega) G_{zz}(\boldsymbol{x}_2, \boldsymbol{\xi}, \omega) f_{zz}(\boldsymbol{x}_1, \boldsymbol{x}_2, \omega)\, d\omega, \tag{8}$$

where merely the function $f_{zz}$ is determined by the chosen misfit function and corresponds to the derivative of the misfit function with respect to the modelled cross-correlation.

As first misfit function, we use the L2-norm of the synthetic ($\mathcal{C}^{\text{syn}}$) and observed ($\mathcal{C}^{\text{obs}}$) correlation waveforms, i.e.

$$\chi_{\text{fwi}} = \frac{1}{2}\left[\mathcal{C}^{\text{syn}} - \mathcal{C}^{\text{obs}}\right]^2 \tag{9}$$

in time domain, yielding

$$f(\boldsymbol{x}_1, \boldsymbol{x}_2, \omega) = \mathcal{F}\left[\mathcal{C}^{\text{syn}} - \mathcal{C}^{\text{obs}}\right], \tag{10}$$

where we denote the Fourier transform by $\mathcal{F}$.

An exemplary waveform sensitivity kernel for the z-components of both receivers, and vertical sources, is shown in the left panel of Figure 5. It reveals how various locations of the source distribution affect the measurement. One can clearly recognize the pattern of stationary phase regions behind the stations and the oscillating sensitivity in between the stations (e.g. Snieder, 2004; Xu et al., 2019).

In contrast, the right panel of Figure 5 shows sensitivity $K_{zz}$ of another misfit function,

$$\chi_{\text{A}} = \frac{1}{2}\left[A(\mathcal{C}^{syn}) - A(\mathcal{C}^{obs})\right]^2, \tag{11}$$

where

$$A(x(\tau)) = \ln\left(\frac{\int [w_+(\tau)x(\tau)]^2 d\tau}{\int [w_-(\tau)x(\tau)]^2 d\tau}\right), \tag{12}$$

and $w_+, w_-$ denote causal and a-causal window of the cross-correlation, respectively, and $f$ becomes (where the dependency on the lag $\tau$ is omitted):

$$f(\boldsymbol{x}_1, \boldsymbol{x}_2, \omega) = \mathcal{F}\left[\left[A^{\text{syn}} - A^{\text{obs}}\right] \cdot \left[\frac{w_+^2 \mathcal{C}^{\text{syn}}}{\int [w_+ \mathcal{C}^{\text{syn}}]^2 d\tau} - \frac{w_-^2 \mathcal{C}^{\text{syn}}}{\int [w_- \mathcal{C}^{\text{syn}}]^2 d\tau}\right]\right]. \tag{13}$$

For simplicity, we will refer to this second measurement as asymmetry in the following. This second sensitivity kernel (Figure 5, right panel) is smoother than the full-waveform one: The oscillating sensitivity between the stations is removed due to the windowing by $w_-, w_+$, and the stationary phase regions have opposite signs of sensitivity due to the ratio $\frac{\int [w_+(\tau)x(\tau)]^2 d\tau}{\int [w_-(\tau)x(\tau)]^2 d\tau}$. A body wave is caught in the measurement window, adding a faint ring of sensitivity near the stations probably due to body-wave surface-wave interaction (Sager et al., 2018a). The term $f_{zz}(\boldsymbol{x}_1, \boldsymbol{x}_2, \omega)$ encompasses the differences between both sensitivity kernels of Figure 5, by taking the form of equations 10 and 13 for waveforms and asymmetry measurement, respectively.

This illustrate that inversions using different strategies to measure data-model misfit (waveform, asymmetry, etc.) will produce different optimal models of the noise source distribution. For example, provided adequate coverage, one can expect a higher resolution to result from using the L2 waveform misfit, which has more short-wavelength spatial features. This appears even more clearly once we conduct the inversion. We first construct a synthetic dataset by forward modelling cross-correlations from a source distribution shown in Figure 6, upper left panel, which has a low background level of sources in the left half of the domain, along with three strong Gaussian-shaped sources, marked by green crosses, at varying distance outside the array, which is marked by red triangles. The right half of the domain is source-free. The frequency content of the starting model is homogeneous for all sources (background and blobs), with Gaussian power spectral density $S(\boldsymbol{\xi}, \omega)$ of equation 3 having a mean frequency 0.05 Hz and standard deviation of 0.02 Hz. We compute cross-correlations through PREM at all stationpairs of the array, and add Gaussian noise with an amplitude of $\pm 5\%$ of the average root mean square of all synthetic cross-correlation traces.

To treat the inversions with different measurements consistently, we proceed in the same manner concerning filtering and smoothing. The inversion starts at lower frequency, and a higher frequency band is added (taking two measurements after bandpass filtering in two different bands) after 20 iterations. Gaussian smoothing is applied in lieu of a more formal regularization, and smoothing length is decreased after 20 iterations. The optimization itself is performed with the L-BFGS algorithm of the scipy minimize module (Nocedal and Wright, 2006; Millman and Aivazis, 2011). Results are shown in Figure 6. The second row shows results from full waveform inversion (left panel) and asymmetry inversion (right panel). The centers of the Gaussian perturbations to be retrieved are marked by green crosses also on the recovered models, to simplify comparison with the target model. Titles indicate the respective measurements and numbers in brackets show the minimum and maximum of the recovered source distributions; the maximum amplitude of 1 is not fully recovered by any of the inversions, due to the smoothing regularization.

As expected, the full waveform misfit performs better at recovering the perturbations. The recovery succeeds reasonably well for sources that are close to the array, whereas sources at greater distance are more smeared both towards and away from the array. The sources close to the array suffer fairly little smoothing, and demonstrate that it is possible to not only retrieve the direction, but in this case also the approximate location of ambient noise sources predominantly imaged by fitting surface wave measurements.

The logarithmic signal energy ratio misfit shows stronger inversion artifacts and images a rather crude impression of the target model, with stronger smearing effects. In addition, this inversion was terminated after 44 iterations due to falling below the threshold for minimal misfit improvement, which might indicate that it is trapped in a local minimum, or simply suggests very

slow convergence.

The bottom of Figure 6 shows example waveforms for two station pairs. Predicted waveforms by the final models (blue lines) are shown along with noise-free synthetic data (dark gray) and the synthetic data with additive noise which were used for inversion (light gray). Note that the gray traces do not vary between left and right column, whereas the blue traces show results for different measurements. Traces in the first row correspond to a station pair which is oriented Southwest-Northeast and marked by dashed circles, i.e. its stationary phase aligns approximately with the source at $(-3°, -3°)$: the signal-to-noise-ratio is high, and the waveform measurement results in an excellent fit to the noise-free synthetic data. On the other hand, the bottom row corresponds to a station-pair oriented North-South, marked by solid circles. In this case, sources in the stationary phase region are very low, and strong sources are located outside of it. The signal-to-noise ratio is low and the fit worse, with some degree of overfitting. The asymmetry measurement appears to be more sensitive to additive noise, and performs worse at recovering waveforms. For the favorably oriented station pair, it recovers phases resonably well; amplitudes cannot be recovered with this measurement because it is based on a ratio that removes absolute amplitude information. For the unfavourably oriented station pair, neither phase nor amplitude fit well.

While the full waveform misfit produces a very satisfactory image in this synthetic case, it has very low tolerance to errors in the seismic velocity model (Sager et al., 2018b; Xu et al., 2019). On the other hand, the logarithmic energy ratio misfit, which produces a poor image of the target, is very robust with respect to perturbations of the velocity model (Sager et al., 2018b) and has been shown to perform better in scenarios with spatially separated source perturbations (Ermert et al., 2017; Sager et al., 2018b). Our proposed strategy for ambient seismic source inversion is to consider several misfits for inversion and base interpretations on the synopsis of the results. The modular structure of `noisi` allows to implement new measurement functions without adapting any other part of the code by adding functions with the same call and return parameters to these scripts. Besides the measurements illustrated above, the L2-misfit of signal energy in the surface wave window is implemented.

## 6   Discussion and conclusions

The `noisi` tool allows users to create correlations for a variety of source models without the burden of costly numerical wave propagation simulations by utilizing `instaseis`, or to run noise source inversion at reduced cost with pre-calculated Green's function databases from AxiSEM3D or other wave propagation solvers. Due to its implementation in Python, the tool can be easily modified and integrated into a rapidly growing ecosystem of seismologic applications in Python (e.g. Beyreuther et al., 2010; Hosseini and Sigloch, 2017; van Driel et al., 2015a; Heimann et al., 2019; Lecocq et al., 2014).

Disadvantages compared to implementations integrated into spectral element solvers, such as the ones by Tromp et al. (2010) and Sager et al. (2018a), is the rigid setting of the source grid, and the approximation of spatial integrals. These are evaluated by weighted sums, which can lead to approximation artifacts (see Figure 2). Tromp et al. (2010); Basini et al. (2013) and Sager et al. (2018a) evaluate the spatial integrals using the spectral element basis, which is expected to approximate the integral better at comparable spatial resolution. However, this is not a conceptual but rather a current practical limitation of the tool, and could thus be overcome by adapting the wavefield storage and spatial integration. While the errors in Figure 2, bottom panels, may

appear large, they may often be negligible in comparison to data noise, and can be further diminished by increased spatial sampling. The storage burden of the Green's function database may be regarded as another disadvantage. However, wavefields at the surface of the modeling domain have to be temporarily stored in either type of implementation to allow the application of the ambient source spectra, and thus the choice to re-use them appears intuitive. Finally, and most importantly, the tool is not fit to perform ambient noise full waveform adjoint tomography. This task requires iterative updates to the Earth model, and can be achieved by SPECFEM3D_globe or the recently developed Salvus (Afanasiev et al., 2018). Both of these implement a spectral element model of the cross-correlation wavefield (Tromp et al., 2010; Sager et al., 2020). Extension of `noisi` to compute structure sensitivity kernels is possible but highly impractical, because storing the required volume wavefield would be cumbersome and re-computation of the wavefield after each structural update would defeat the purpose of using pre-computed wavefields.

The output of the wavefield at the Earth's surface either in full or sampled at particular pre-defined grid locations poses practical challenges for input / output and storage in both types of applications. As an example, the retained wavefield utilized by SPECFEM3D_GLOBE for creating the cross-correlations of a single reference station in Figure 2 amounts to 180 GB for the 40 by $40°$ domain with 15 s shortest period. Furthermore, the wavefield at the surface needs to be either post-processed for usage with `noisi`, or convolved with the ambient noise source spectrum (e.g. Tromp et al., 2010). This is made cumbersome by the high temporal sampling of the numerical wavefield, which is imposed by the CFL-criterion. The ease of computing cross-correlations with `noisi` is in part a consequence of decimating simulated Green's functions in time by factors of 10 and more. In turn, built-in sparser representation and / or output of the surface wavefield in numerical solvers, such as currently implemented by Salvus (Boehm et al., 2016) partially alleviates the burden and may pave the way for faster and computationally cheaper noise cross-correlation modeling without recourse to pre-calculated wavefields. In the meantime, further developments of the presented tool may include improvements of the spatial integration. To the best of our knowledge, it closes a current gap in the application of Green's function databases for noise cross-correlation modeling.

*Code and data availability.* The Python code can be downloaded from github (https://github.com/lermert/noisi). A tutorial in the form of a jupyter notebook is provided as the main item of documentation, and details each step for the computation of cross-correlations and sensitivity kernels.

The github repository contains a set of basic test cases to be passed by further developments. It also provides a numerical test for the consistency of forward model and gradient, which can be employed for the development of additional misfit functions.

All observed seismic data used to prepare this manuscript were downloaded from IRIS Data Management Center.

## Appendix A: Sac headers

The following SAC headers on observed cross-correlation traces can be used with `noisi` in order to perform measurements with the goal of ambient seismic source inversion. Only few of them are essential to provide the necessary information to the tool. These are marked in bold:

b: (float), minimum lag

e: (float), maximum lag

**stla**: (float), Latitude of station 1

**stlo**: (float), Longitude of station 1

**evla**: (float), Latitude of station 2

**evlo**: (float), Longitude of station 2

user0: (float), Number of stacked windows

user1: (float), window length for observed ross-correlation computation

user2: (float), window overlap during observed cross-correlation computation

**dist**: (float), station pair distance in m

az: (float), station pair azimuth in degree

baz: (float), station pair back azimuth in degree

**kstnm**: (string), station code of station 1

**kevnm**: (string), station code of station 1

kt0: (string), date of earliest window in cross-correlation stack (YYYYjjj)

kt1: (string), date of latest window in cross-correlation stack (YYYYjjj)

**kuser0**: (string), network code of station 2

kuser1: (string), location code of station 2

**kuser2**: (string), channel code of station 2

**kcmpnm**: (string), channel code of station 1

**knetwk**: (string), network code of station 1

## Appendix B: Example input station list

Stations to be used in modeling need to be specified in a comma-separated list (with one example line):

net,sta,lat,lon

G,CAN,-35.318715,148.996325

## Appendix C: Wavefield format

The tool expects to find Green's functions organized as hdf5 files by seismic receiver channel, with filenames NETWORK.
STATION..CHANNEL.h5 for the networks and stations listed in the input file list (see above). Each hdf5 file needs to contain the following data structure. Both single and double precision floats may be used for the "data" and "sourcegrid" datasets. Single precision is used by default.

group "/"

        dataset "data" (float), shape: ntraces by nt, Green's functions

        dataset "sourcegrid" (float), shape: 2 by ntraces, geographic grid

        dataset "stats", metadata                         attribute "Fs" (float), sampling rate in Hz

                                                          attribute "data_quantity" (string), "DIS", "VEL" or "ACC"

                                                          attribute "fdomain" (int), 0 for time domain, 1 for frequency domain

                                                           attribute "nt" (int), number of samples

                                                           attribute "ntraces" (int), number of source locations

                                                           attribute reference_station (string), SEED identifier of station

## Appendix D: Noise source format

The tool expects to find the noise source model as hdf5 file with name starting_model.h5 (for each iteration) with the following data structure:

group "/"

        dataset "coordinates" (float), shape: 2 by ntraces; geographic grid

        dataset "frequencies" (float), shape: Number of frequency samples after zero-padded, next power of 2, real FFT of nt; frequency axis

        dataset "model" (float), shape: ntraces by number of basis functions; spatial weights of noise source model

        dataset "spectral_basis" (float), shape: number of basis functions by length of frequency axis; spectral basis functions

        dataset "surface_areas" (float), shape: ntraces; approximate surface area of grid cell

*Author contributions.* LE implemented the current version of noisi and ran the comparison with specfem, the forward model and the synthetic inversions. KS and JI continuously contributed suggestions for the conceptual setup and improvement of the tool. KS provided advice on the inversion, and JI intensively test-ran the code in the course of further developments. ES computed the global hum source input model and reviewed the observational and synthetic cross-correlation results. ES, TNM and AF provided scientific advice. A first version of the manuscript was drafted by LE and continuously improved upon by discussions with and suggestions from all authors.

*Competing interests.* The authors declare not to have any competing interests.

*Acknowledgements.* The authors thank Kees Wapenaar and Erdinc Saygin for their thoughtful, constructive reviews on the manuscript, and the topical and executive editors of Solid Earth, Michal Malinowski and Charlotte Krawczyk, for its efficient handling. LE gratefully acknowledges support from the Swiss National Science Foundation under grant P2EZP2_175124, and thanks Naiara Korta Martiartu for improvement suggestions to the manuscript. A warm thank you also to Kuangdai Leng for valuable help with AxiSEM3D. KS thanks the Swiss National Science Foundation for support under grant P2EZP2_184379. Computations were run on the UK National Supercomputer ARCHER under LE's driving test allocation and TNM's NERC remit. Observational data was obtained from the IRIS Data Management Center. IRIS Data Services are funded through the Seismological Facilities for the Advancement of Geoscience and EarthScope (SAGE) Proposal of the National Science Foundation under Cooperative Agreement EAR-1261681.

480

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

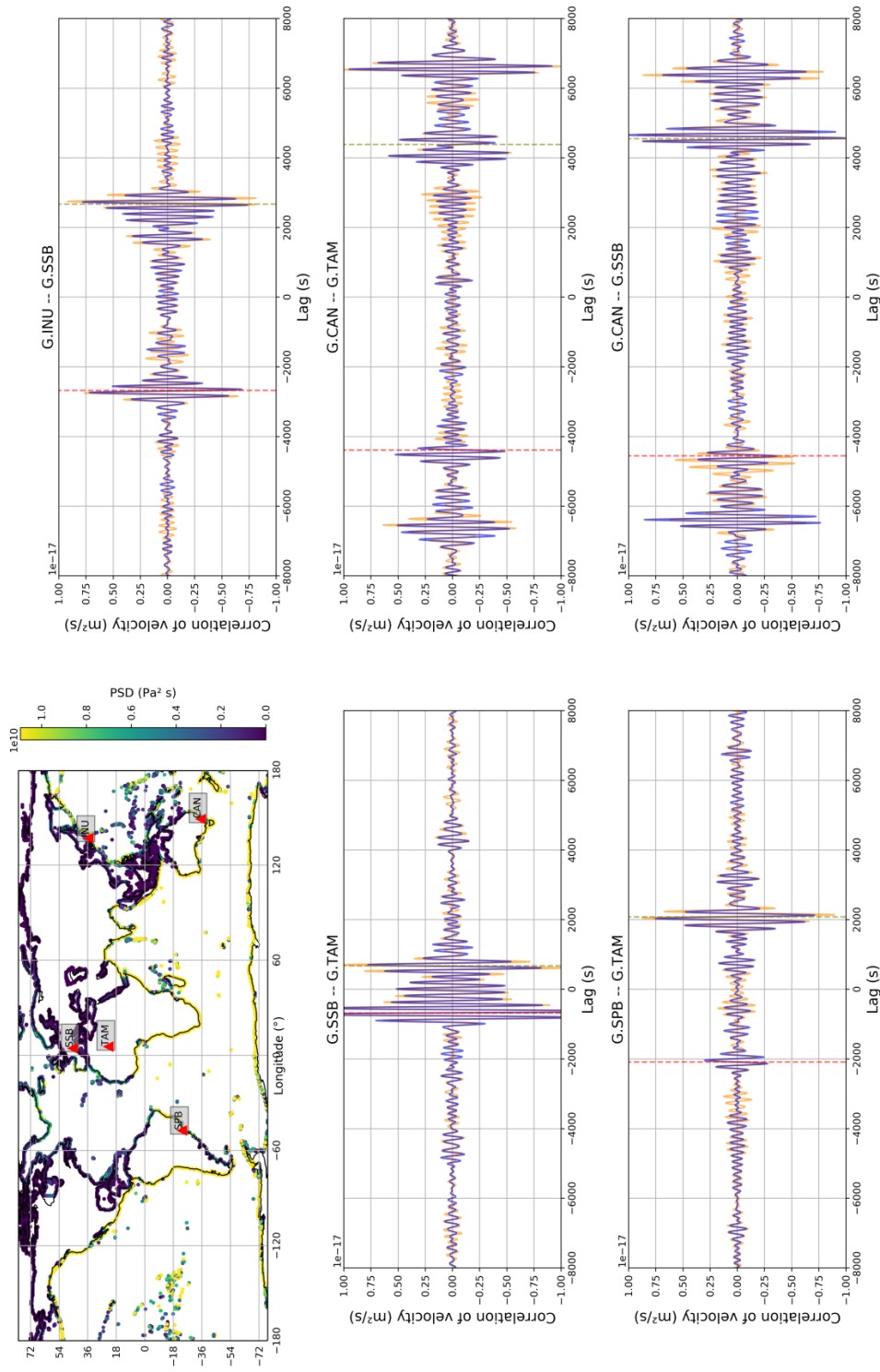

**Figure 3.** Simulation of cross-correlations due to hum sources modeled akin to Ardhuin et al. (2015); Deen et al. (2018). Hum sources are localized in small areas constrained to shorelines of continents and islands. Correlations were computed using both anisotropic PREM (yellow) and S40RTS (blue). Localized hum sources cause a host of early-arriving surface wave phases in the cross-correlations. Red and green vertical lines mark the arrival time of a surface wave travelling at $3.7 \frac{km}{s}$

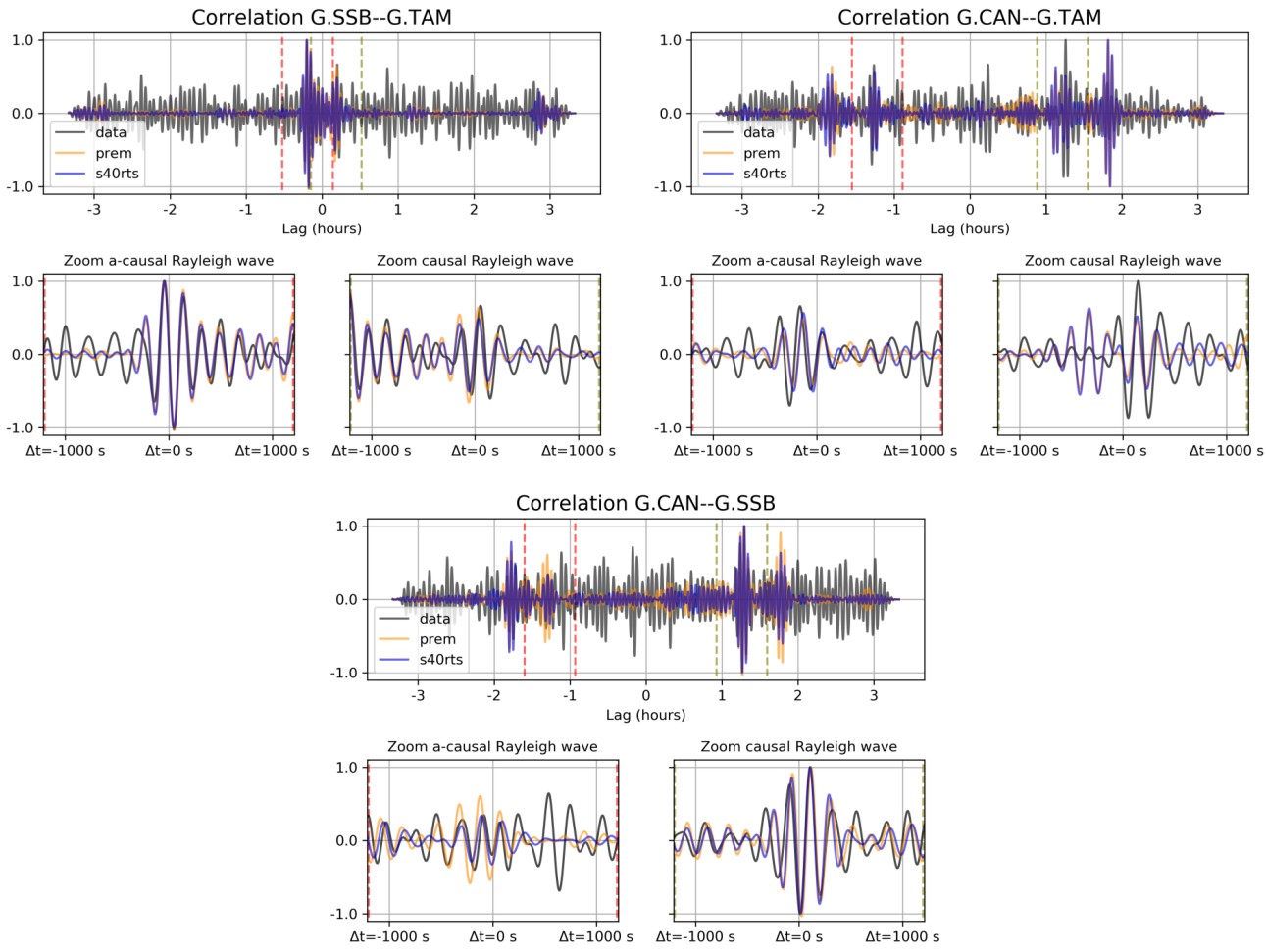

**Figure 4.** Forward modeled and observed cross-correlations. No fitting or inversion was undertaken; the forward model is built upon the hum mechanism by Ardhuin et al. (2015) and Deen et al. (2018), and using PREM (yellow lines) and S40RTS (blue lines). Correlations are normalized by maximum amplitude. Red and green vertical lines indicate windows of $\pm$ 20 minutes around a minor-arc surface wave travelling at $3.7 \frac{\text{km}}{\text{s}}$. These are enlarged in respective bottom panels.

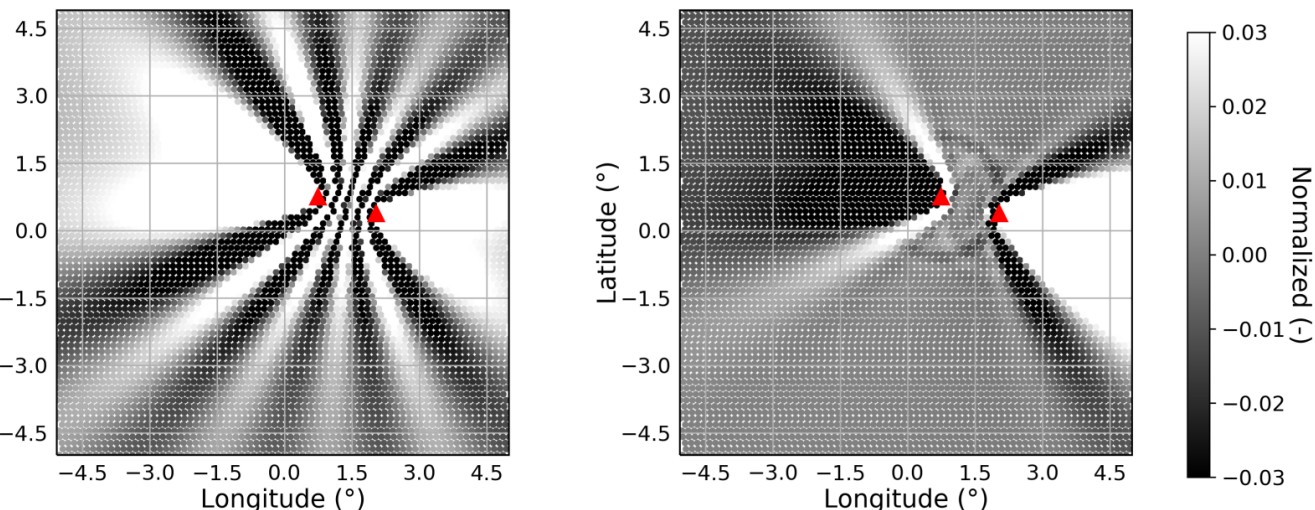

**Figure 5.** Illustration of sensitivity kernels. Left panel: Normalized vertical-component sensitivity kernel $K_{zz}(\boldsymbol{x}_1, \boldsymbol{x}_2, \boldsymbol{\xi})$ of full waveform L2-misfit $\chi_{\mathrm{fwi}}$ (equation 9). The station locations $\boldsymbol{x}_1, \boldsymbol{x}_2$ are marked by red triangles. Frequency integration runs from 0 Hz to the Nyquist frequency, but the source spectrum peaks at dominant frequency 0.05 Hz and filters out everything above 0.1 Hz. Right: Normalized sensitivity kernel $K_{zz}(\boldsymbol{x}_1, \boldsymbol{x}_2, \boldsymbol{\xi})$ of windowed asymmetry measurement $\chi_{\mathrm{A}}$ (equation 11). Similar figures can be obtained by adapting the Jupyter notebook tutorial for noisi.

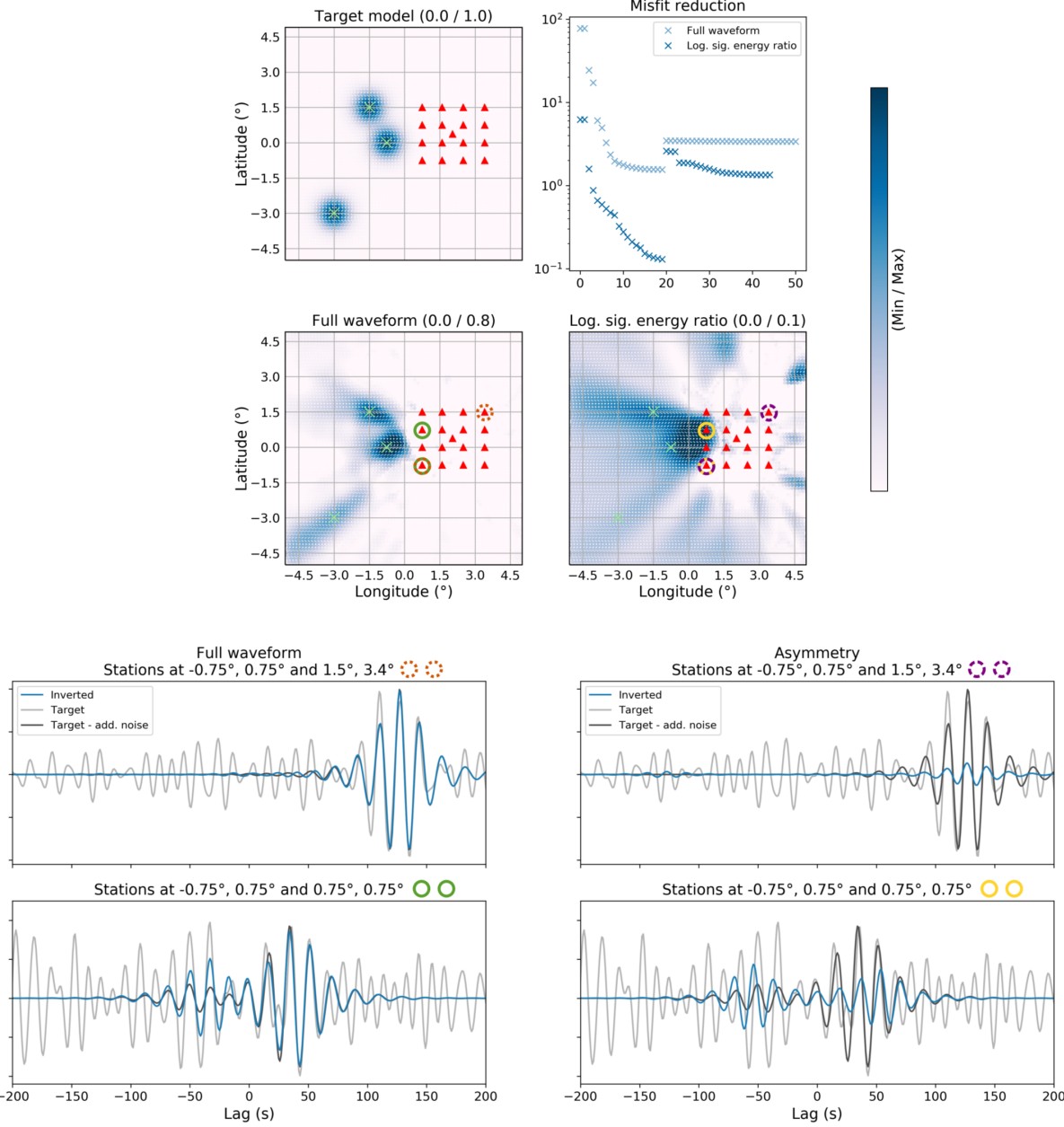

**Figure 6.** Top: Synthetic inversions of ambient seismic source distribution. The target model is shown in the top left panel. The top right panel shows the misfit reduction using two different measurements. After 20 iterations, an additional frequency band was added to the inversion, and smoothing decreased. Center: Recovered source distributions. Titles indicate the respective measurement; the numbers in brackets indicate the minimum and maximum values of the color scale. Bottom: Comparison of waveforms from the final models to the synthetic data. For this comparison, we chose a particularly good (top row) and a particularly bad example (bottom row). Synthetic data from target model, including additive noise, are shown by light gray lines. For comparison, we also show the noise-free synthetics in dark gray lines, which were not used for inversion, but show that the inverted model retrieves the coherent information rather than the random noise. Modelled waveforms obtained from the inverted source distributions based on the full waveform and asymmetry are shown in blue. Colored circles indicate the location of the station pairs.