# Peer review of "noisi: A Python tool for ambient noise cross-correlation modeling and noise source inversion"

_Solid Earth, 2020_

## Referee Comment (RC1) · Cornelis Wapenaar (Referee) · 28 May 2020

This is an interesting paper about a tool for efficient modelling of noise cross-correlations with applications in noise source inversion. Overall the paper is well written, but at places it is can use some additional explanations.

Here are some examples:

It is not clear why the correlation function is first introduced in terms of integrals (eqs 2 and 4) and later discretized (eq 7). Since in this paper the correlation function is not used as a representation of a Green's function between $x\_1$ and $x\_2$, it seems that one doesn't need the integral form of eq 4. Wouldn't it make more sense to define the correlation function as a summation (like in eq 7) right from the start? (honoring the

fact that sources are in general sparsely and irregularly distributed, such as in Figures 3a and 6a). If there is a good reason to start with the integral representation then this should be clearly explained.

Figure 5 shows sensitivity kernels. This needs more explanation. I assume the left panel shows eq 6 as a function of xi, for fixed x_1 and x_2 (shown by the triangles) and fixed n and m. If this is correct, please state this explicitly (or if it is not correct, explain what it shows instead). How are omega_0 and omega_1 chosen? The right panel, which shows the function A of eq 8, needs even more explanation. There are no spatial variables in eq 8. I assume C in eq 8 is implicitly a function of x_1 and x_2, but not of xi. It remains unclear to me which of these variables are taken along the axes in the right panel of Figure 5.

Some minor points:

Lines 15-20: I suggest to add some references to the pioneering papers in each of the fields of application mentioned here.

Line 51: 'It presents a . . . alternative for cross-correlation modeling.' Why do you say 'an alternative'? Isn't cross-correlation modeling what the paper is about?

Line 93: The star instead of the bracket should be set as a superscript.

---

## Referee Comment (RC2) · Erdinc Saygin (Referee) · 1 Jun 2020

This work aims to fill a significant information gap in the field of seismic ambient noise. Authors did an excellent job in articulating the importance and capabilities of the "noisi" package.

Therefore, my comments are mainly stylistic in nature.

-Lines 30-, I am not sure if DSurfTomo will qualify for ambient noise data processing tool.

-Please introduce "Syngine" as "The IRIS Synthetics Engine" at the beginning.

-For equations 1 and 2, convolution and correlation terms are used in the text. However,

the equations itself are in the frequency domain and are in multiplicative form.

-Equation 3 has a minor typo.

-Page 9, Line 240 is vague, and itself does not make much sense.

-Figure 3 caption has autocorrelation, but all of the waveforms are from cross-correlations.

-Section 5.2: Are the sources in Figure 6 simultaneously or randomly occurring. Is there any delay? And what is the frequency content?

-Between 345 and 350 "wave forms" should be "waveforms".

Figure 6: Rather than giving coordinates, please mark/number the selected stations and use them in the waveform plots

I found the use of logarithmic signal ratio and asymmetry inversion confusing. Can you please describe it further or rephrase the part in Lines 310 onwards.

---

## Author Comment (AC1) · 23 Jun 2020

Thank you very much for your review! I will upload a file below as reply. The file will contain answers to your comments, and outline some changes that we have made to the manuscript. Uploading the revised manuscript may be not be possible yet, only after the public discussion is closed; in any case I hope that our reply will detail clearly enough how we changed the manuscript. Thank you again on behalf of all the authors.

Please also note the supplement to this comment:
https://se.copernicus.org/preprints/se-2020-57/se-2020-57-AC1-supplement.pdf
* * *
[Figure]

**Supplement:**

**Referee 1 (Cornelis Wapenaar):**

*It is not clear why the correlation function is first introduced in terms of integrals (eqs 2 and 4) and later discretized (eq 7). Since in this paper the correlation function is not used as a representation of a Green's function between x_1 and x_2, it seems that one doesn't need the integral form of eq 4. Wouldn't it make more sense to define the correlation function as a summation (like in eq 7) right from the start? (honoring the fact that sources are in general sparsely and irregularly distributed, such as in Figures 3a and 6a). If there is a good reason to start with the integral representation then this should be clearly explained.*

We considered this question in detail and discussed about it. We agree with the reviewer that the choice to start from an integral representation is somewhat arbitrary because we later approximate by summation.

The reason why we would prefer to continue using the integral representation is that for the most general case, we assume that noise excitation can be represented by sources acting and varying continuously over extended areas. While we don't assume that their occurrence is homogeneous, we believe that for example the secondary microseism is excited by sources distributed over areas where strong, opposing wave trains occur, for example through storms, and which are not always strongly localized.

In this sense, the example in Figure 3 is not the most representative case. Sources in Figure 6 are actually continuously distributed, although most of them have low amplitudes compared to the Gaussian-shaped, more localized ones.

We added the following sentence to make this thought process explicit:
"We adopt an integral description here, as we assume that the noise sources $N(\xi_1, w)$ and $N(\xi_2, w)$ are generally extended and vary continuously over more or less extended source areas."

*Figure 5 shows sensitivity kernels. This needs more explanation. I assume the left panel shows eq 6 as a function of xi, for fixed x_1 and x_2 (shown by the triangles) and fixed n and m. If this is correct, please state this explicitly (or if it is not correct, explain what it shows instead). How are omega_0 and omega_1 chosen?*

Indeed the left panels shows equation 6. We have added the suggested information, which indeed helps clarifying the figure caption, namely that station locations $x_1$, $x_2$ are marked by triangles and that nm are fixed to zz. Integration of omega is over the entire frequency band that was used in the simulation, but the source spectrum, which peaks at 0.05 Hz in this case, acts as a filter. (See also the point below).

*The right panel, which shows the function A of eq 8, needs even more explanation. There are no spatial variables in eq 8. I assume C in eq 8 is implicitly a function of x_1 and x_2, but not of xi. It remains unclear to me which of these variables are taken along the axes in the right panel of Figure 5.*

What is shown is actually not A, but the sensitivity kernel that one obtains after measuring A on both synthetic and observed cross-correlations. This was apparently wholly unclear from our description and the figure caption, since the usage of the measurement also led to questions from referee 2. Therefore, we have adapted the respective paragraph (lines 310 ff in the original manuscript) and mentioned which functions are plotted in both panels. The figure caption now reads:
"Illustration of sensitivity kernels. Left panel: Normalized vertical-component sensitivity kernel $K_{zz}$ ($x_1$, $x_2$, $\xi$) of full waveform L2-misfit $\chi_{fwi}$ (equation 9). The station locations $x_1$, $x_2$ are marked by red triangles. Frequency integration runs from 0 Hz to the Nyquist frequency, but the source

spectrum peaks at dominant frequency 0.05 Hz and filters out everything above 0.1 Hz. Right: Normalized sensitivity kernel $K_{zz}(x_1, x_2, \xi)$ of windowed asymmetry measurement $\chi_A$ (equation 11). Similar figures can be obtained by adapting the Jupyter notebook tutorial for noisi."

(Note that equation numbers have shifted with respect to the original manuscript). Please also consider the answer to the last question of Referee 2, where we incorporate changes of the mentioned paragraph.

***Lines 15-20: I suggest to add some references to the pioneering papers in each of the fields of application mentioned here.***
We understand this suggestion well. Since there is such a large number of high-quality publications in the field, our selection here is somewhat arbitrary, and citing the pioneering papers is more systematic. We have added references, and the paragraph now reads:

"Cross-correlations of ambient seismic noise form the basis of many applications in seismology, from site effects studies (e.g., Aki, 1957; Roten et al., 2006; Bard et al., 2010; Denolle et al., 2013; Bowden et al., 2015) to ambient noise tomography (e.g., Shapiro et al., 2005; Yang et al., 2007; Nishida et al., 2009; Haned et al., 2016; de Ridder et al., 2014; Fang et al., 2015; Singer et al., 2017) and coda wave interferometry (e.g., Sens-Schönfelder and Wegler, 2006; Brenguier et al., 2008; Obermann et al., 2013; Sánchez-Pastor et al.,2019). Auto-correlations of the ambient noise are also increasingly used to study seismic interfaces as suggested by Claerbout (1968) (e.g., Taylor et al., 2016; Saygin et al., 2017; Romero and Schimmel, 2018) and to monitor subsurface properties (Viens et al., 2018; Clements and Denolle, 2018)."

***Line 51: 'It presents a . . . alternative for cross-correlation modeling.' Why do you say 'an alternative'? Isn't cross-correlation modeling what the paper is about?***
What we mean here is that the tool is an alternative approach to modeling cross-correlations compared to the implementations in specfem3d and salvus, which are more versatile, but also more computationally expensive. To make this clearer, we adapted the paragraph as:

"The noise cross-correlation implementations of Tromp et al. (2010) and Sager et al. (2018a) honor the physics of wave propagation to the greatest possible extent, but require substantial HPC resources for inversion (Sager et al., 2020). The noisi tool uses databases of pre-calculated seismic wavefields instead to compute cross-correlations and sensitivity kernels. It therefore presents a computationally inexpensive alternative for cross-correlation modeling and noise source inversion when updates to the structure model (i.e., seismic velocities, density, and attenuation) are not required. Owing to the reuse of Green's functions, computation is quick and inexpensive. However, storage resources, typically on the order of 1 GB per station, are needed to hold the Green's function database."

***Line 93: The star instead of the bracket should be set as a superscript.***
This has been corrected.

---

## Author Comment (AC2) · 23 Jun 2020

Thank you for the review! We have considered your comments, and I will append a file (supplement.....pdf) below which details how we have revised the manuscript to address them. The largest change is with regard to the inversion using the asymmetry measurement, where we included a new paragraph to improve its description. Best regards from all of us.

Please also note the supplement to this comment:
https://se.copernicus.org/preprints/se-2020-57/se-2020-57-AC2-supplement.pdf

**Supplement:**

**Referee 2 (Erdinc Saygin):**

*Lines 30-, I am not sure if DSurfTomo will qualify for ambient noise data processing tool.*
We agree with the reviewer that DSurfTomo is a tool for tomography and thus does not fit into the suite of packages mentioned here. Originally, the manuscript contained also references to packages for ambient noise tomography, which we have since removed for brevity and clarity. Thus, we removed the mention of DSurfTomo to be more consistent.

*Please introduce "Syngine" as "The IRIS Synthetics Engine" at the beginning.*
We've adapted the text accordingly and added the DOI of the IRIS repository as follows:
"This is achieved, for example, by the IRIS Synthetics engine (Syngine) repository (IRIS, 2015; Krischer et al., 2017) and by tools for the extraction and management of Green's function databases (van Driel et al., 2015a; Heimann et al., 2019)."

*For equations 1 and 2, convolution and correlation terms are used in the text. However, the equations itself are in the frequency domain and are in multiplicative form.*
This could indeed be misunderstood. We have added explicit mention of the change to frequency domain to both the relevant sentences. They now read:
"One component of ground motion u i observed at a seismic receiver at location x can be modeled as the convolution of the noise source time series with the impulse response of the Earth or Green's function G. In frequency domain, this relation is expressed as

$$u_i(\boldsymbol{x},\omega) = \int_{\partial\oplus} G_{in}(\boldsymbol{x},\boldsymbol{\xi},\omega)N_n(\boldsymbol{\xi},\omega)\,\mathrm{d}\boldsymbol{\xi}^2,$$

(Aki and Richards, 2002), where summation over repeated indices is implied. The correlation of two such signals, averaged over an observation period, can be expressed by multiplication in the frequency domain, i.e.

$$\mathcal{C}_{ij}(\boldsymbol{x}_1,\boldsymbol{x}_2,\omega) = \langle u_i^*(\boldsymbol{x}_1,\omega)u_j(\boldsymbol{x}_2,\omega)\rangle$$
$$= \left\langle \iint_{\delta\oplus} G_{in}^*(\boldsymbol{x}_1,\boldsymbol{\xi}_1,\omega)N_n^*(\boldsymbol{\xi}_1,\omega)G_{jm}(\boldsymbol{x}_2,\boldsymbol{\xi}_2,\omega)N_m(\boldsymbol{\xi}_2,\omega)\,\mathrm{d}\boldsymbol{\xi}_1\mathrm{d}\boldsymbol{\xi}_2 \right\rangle,$$

etc.

*Equation 3 has a minor typo.*
Thank you, the typo was corrected.

*Page 9, Line 240 is vague, and itself does not make much sense.*
The sentence in question stated: "The simulation in AxiSEM3D is run for a global mesh, but dropping the number of Fourier expansion coefficients to 0 at a depth of 800 km and at a distance above 90 degree." We understand that more detail is required to make the sentence less vague. We suggest to replace the sentence by the following short paragraph:

"In AxiSEM3D, a method that couples a spectral-element discretisation with a pseudospectral expansion along the azimuth (Leng et al., 2019) we simulate the full desired 3D resolution inside the domain of interest. Rather than using absorbing boundaries as in the simulation with SPECFEM3D\ _GLOBE, we avoid spurious reflections in AxiSEM3D by using a global computational domain. The azimuthal Fourier expansion is tapered to a minimum of two Fourier coefficients outside of our domain of interest, which strongly reduces the additional compute time accrued due to the global simulation."

Note that we used this particular approach because absorbing boundaries didn't ship with the published version of AxiSEM3D at the time the simulations were conducted. Their development was ongoing at the time and is soon to be published by Haindl et al.

*Figure 3 caption has autocorrelation, but all of the waveforms are from cross-correlations.*
This has been corrected.

**Section 5.2: Are the sources in Figure 6 simultaneously or randomly occurring. Is there any delay? And what is the frequency content?**
The sources are assumed to have converged to the expected value of a random, uncorrelated field of sources as expressed in equation 3. Equation 3 yields power spectral density estimate Snm(xi, omega) as source term, i.e. phase information is no longer contained in the source term. This assumption may be questioned, but it is pervasively used in ambient noise studies, both of the Green's function retrieval and the cross-correlation modeling kind.

We propose to extend the sentence in lines 325-326 as follows:
"The frequency content of the starting model is homogeneous for all sources (background and blobs), with Gaussian power spectral density S(xi, omega) of equation 3 having a mean frequency 0.05 Hz and standard deviation of 0.02 Hz."

We did not modify the figure caption itself, as it is already a little bit lengthy.

*Between 345 and 350 "wave forms" should be "waveforms".*
Thank you, this has been corrected.

*Figure 6: Rather than giving coordinates, please mark/number the selected stations and use them in the waveform plots.*
This is an excellent suggestion, and we have added a marker for the station pairs in question.

*I found the use of logarithmic signal ratio and asymmetry inversion confusing. Can you please describe it further or rephrase the part in Lines 310 onwards.*
Since Referee 1 also commented on this part, we have attempted to revise the paragraph describing the respective sensitivity kernels in lines 310 ff thoroughly, and we have included the equations describing both misfit functions (full waveform and asymmetry). We hope that this clarifies the usage of the asymmetry in the inversion. The paragraph now reads:

"Sensitivity kernels computed with noisi can be used to run gradient-based inversion for the distribution of ambient seismic sources from a data set of observed ambient noise cross-correlations. To demonstrate the effectiveness of this approach, we conduct two synthetic inversions using two different functions to measure the misfit between observations and model. The sensitivity kernel of any misfit function can be expressed as

$$K_{zz}(\boldsymbol{x}_1,\boldsymbol{x}_2,\boldsymbol{\xi}) = \int_{\omega=0}^{\omega_{Nyq}} G_{zz}^*(\boldsymbol{x}_1,\boldsymbol{\xi},\omega) G_{zz}(\boldsymbol{x}_2,\boldsymbol{\xi},\omega) f_{zz}(\boldsymbol{x}_1,\boldsymbol{x}_2,\omega)\, d\omega,$$

where merely the function f zz is determined by the chosen misfit function and corresponds to the derivative of the misfit function with respect to the modelled cross-correlation.
As first misfit function, we use the L2-norm of the synthetic (C syn ) and observed (C obs ) correlation waveforms, i.e.

$$\chi_{\text{fwi}} = \frac{1}{2}\left[\mathcal{C}^{syn} - \mathcal{C}^{obs}\right]^2.$$

in time domain, yielding

$$f(\boldsymbol{x}_1,\boldsymbol{x}_2,\omega) = \mathcal{F}\left[\mathcal{C}^{\text{syn}} - \mathcal{C}^{\text{obs}}\right],$$

where we denote the Fourier
transform by F. An exemplary waveform sensitivity kernel for the z-components of both receivers, and vertical sources, is shown in the left panel of Figure 5. It reveals how various locations of the source distribution affect the measurement. One can clearly recognize the pattern of stationary phase regions behind the stations and the oscillating sensitivity in between the stations (e.g. Snieder, 2004; Xu et al., 2019).
In contrast, the right panel of Figure 5 shows sensitivity K zz of another misfit function,

$$\chi_A = \frac{1}{2}\left[A(\mathcal{C}^{syn}) - A(\mathcal{C}^{obs})\right]^2,$$

where

$$A(x(\tau)) = \ln\left(\frac{\int [w_+(\tau)x(\tau)]^2 d\tau}{\int [w_-(\tau)x(\tau)]^2 d\tau}\right),$$

and w + , w − denote causal and a-causal window of the cross-correlation, respectively, and f becomes (where the dependency on the lag τ is omitted):

$$f(\boldsymbol{x}_1,\boldsymbol{x}_2,\omega) = \mathcal{F}\left[\left[A^{\text{syn}} - A^{\text{obs}}\right] \cdot \left[\frac{w_+^2 \mathcal{C}^{\text{syn}}}{\int [w_+\mathcal{C}^{\text{syn}}]^2 d\tau} - \frac{w_-^2 \mathcal{C}^{\text{syn}}}{\int [w_-\mathcal{C}^{\text{syn}}]^2 d\tau}\right]\right].$$

For simplicity, we will refer to this second measurement as asymmetry in the following. This second sensitivity kernel (Figure 5, right panel) is smoother than the full-waveform one: The oscillating

sensitivity between the stations is removed due to the windowing by $w-$, $w+$, and the stationary phase regions have opposite signs of sensitivity due to the ratio

$$\frac{\int [w_+(\tau)C(\tau)]^2\, d\tau}{\int [w_-(\tau)C(\tau)]^2\, d\tau}.$$

A body wave is caught in the measurement window, adding a faint ring of sensitivity near the stations probably due to body-wave surface-wave interaction (Sager et al., 2018a). The term $f_{zz}(x_1, x_2, \omega)$ encompasses the differences between both sensitivity kernels of Figure 5, by taking the form of equations 10 and 13 for waveforms and asymmetry measurement, respectively."

---

## Author Response (AR2)

**Referee 1 (Cornelis Wapenaar):**

*It is not clear why the correlation function is first introduced in terms of integrals (eqs 2 and 4) and later discretized (eq 7). Since in this paper the correlation function is not used as a representation of a Green's function between x_1 and x_2, it seems that one doesn't need the integral form of eq 4. Wouldn't it make more sense to define the correlation function as a summation (like in eq 7) right from the start? (honoring the fact that sources are in general sparsely and irregularly distributed, such as in Figures 3a and 6a). If there is a good reason to start with the integral representation then this should be clearly explained.*

We considered this question in detail, because it caught us somewhat off guard. The main reason for choosing the integral representation is that we believe that in general, the ambient noise excitation is best represented as continuously extended over some area, and we use discretization merely to approximate the integral description. In this sense, the example in Figure 3 is not the most representative case. Sources in Figure 6 are actually continuously distributed, although most of them have low amplitudes compared to the Gaussian-shaped, more localized ones. We are open to the possibility of ambient noise sources being generally sparse in nature, but we believe that this is not the way that the ambient noise community is currently thinking about them (except, maybe, for the sources of hum which according to Ardhuin et al. (2015) should be confined to shelf breaks and in that case can be considered localized relative to the wavelength of the seismic waves that they excite). We added the following sentence to make this thought process explicit:

"We adopt an integral description here, as we assume that the noise sources $N(\xi_1, w)$ and $N(\xi_2, w)$ are generally extended and continuously varying at least over some source area."

*Figure 5 shows sensitivity kernels. This needs more explanation. I assume the left panel shows eq 6 as a function of xi, for fixed x_1 and x_2 (shown by the triangles) and fixed n and m. If this is correct, please state this explicitly (or if it is not correct, explain what it shows instead). How are omega_0 and omega_1 chosen?*

This is correct, and we have added the suggested points, which indeed help to clarify the figure, namely that station locations $x_1$, $x_2$ are marked by triangles and that nm are fixed to zz. Integration of omega is over the entire frequency band that was used in the simulation.

*The right panel, which shows the function A of eq 8, needs even more explanation. There are no spatial variables in eq 8. I assume C in eq 8 is implicitly a function of x_1 and x_2, but not of xi. It remains unclear to me which of these variables are taken along the axes in the right panel of Figure 5.*

What is shown is actually not A, but the sensitivity kernel that one obtains after measuring A on both synthetic and observed cross-correlations. It appears that this was not clear from our description and the figure caption. The usage of the A measurement also led to questions from referee 2. Therefore, we have adapted the respective paragraph (lines 310 ff in the original manuscript) and mentioned which functions are plotted in both panels. The figure caption now reads:

"Figure 5. Illustration of sensitivity kernels. Left panel: Normalized vertical-component sensitivity kernel $K_{zz}(x_1, x_2, \xi)$ of full waveform L2-misfit $\chi\_fwi$ (equation 8). The station locations $x_1$, $x_2$ are marked by red triangles, and frequency integration is from 0 Hz to the Nyquist frequency. Right: Normalized sensitivity kernel $K_{zz}(x_1, x_2, \xi)$ of windowed asymmetry measurement $\chi\_A$ (equation 10). Similar figures can be obtained by adapting the Jupyter notebook tutorial for noisi."

(Note that equation numbers have shifted). Please also consider the answer to the last question of Referee 2, where we incorporate changes of the mentioned paragraph.

***Lines 15-20: I suggest to add some references to the pioneering papers in each of the fields of application mentioned here.***

We understand this suggestion well. Since there is such a large number of high-quality publications in the field, our selection here is somewhat arbitrary, and citing the pioneering papers is more systematic. We have added references, and the paragraph now reads:

"Cross-correlations of ambient seismic noise form the basis of many applications in seismology, from site effects studies (e.g., Aki, 1957; Roten et al., 2006; Bard et al., 2010; Denolle et al., 2013; Bowden et al., 2015) to ambient noise tomography (e.g., Shapiro et al., 2005; Yang et al., 2007; de Ridder et al., 2014; Fang et al., 2015; Singer et al., 2017) and coda wave interferometry (e.g., Sens-Schönfelder and Wegler, 2006; Brenguier et al., 2008; Obermann et al., 2013; Sánchez-Pastor et al.,2019). Auto-correlations of the ambient noise are also increasingly used to study seismic interfaces as suggested by Claerbout (1968) (e.g., Taylor et al., 2016; Saygin et al., 2017; Romero and Schimmel, 2018) and to monitor subsurface properties (Viens et al., 2018; Clements and Denolle, 2018)."

***Line 51: 'It presents a . . . alternative for cross-correlation modeling.' Why do you say 'an alternative'? Isn't cross-correlation modeling what the paper is about?***

What we mean here is that the tool is an alternative approach to modeling cross-correlations compared to the implementations in specfem3d and salvus, which are more versatile, but also more computationally expensive. To make this clearer, we adapted the paragraph as:

"The noise cross-correlation implementations of Tromp et al. (2010) and Sager et al. (2018a) honor the physics of wave propagation to the greatest possible extent, but require substantial HPC resources for inversion (Sager et al., 2020). The noisi tool uses databases of pre-calculated seismic wavefields instead to compute cross-correlations and sensitivity kernels. It therefore presents a computationally inexpensive alternative for cross-correlation modeling and noise source inversion when updates to the structure model (i.e., seismic velocities, density, and attenuation) are not required."

***Line 93: The star instead of the bracket should be set as a superscript.***

This has been corrected.

**Referee 2 (Erdinc Saygin):**

***Lines 30-, I am not sure if DSurfTomo will qualify for ambient noise data processing tool.***

We agree with the reviewer that DSurfTomo is a tool for tomography and thus does not fit into the suite of packages mentioned here. Originally, the manuscript contained also references to packages for ambient noise tomography, which we have since removed for brevity and clarity. Thus, we removed the mention of DSurfTomo to be more consistent.

***Please introduce "Syngine" as "The IRIS Synthetics Engine" at the beginning.***

We've adapted the text accordingly and added the DOI of the IRIS repository as follows:

"This is achieved, for example, by the IRIS Synthetics engine (Syngine) repository (IRIS, 2015; Krischer et al., 2017) and by tools for the extraction and management of Green's function databases (van Driel et al., 2015a; Heimann et al., 2019)."

***For equations 1 and 2, convolution and correlation terms are used in the text. However, the equations itself are in the frequency domain and are in multiplicative form.***
This could indeed be misunderstood. We have added explicit mention of the change to frequency domain to both the relevant sentences. They now read:
"One component of ground motion u i observed at a seismic receiver at location x can be modeled as the convolution of the noise source time series with the impulse response of the Earth or Green's function G. In frequency domain, this relation is expressed as

$$u_i(\boldsymbol{x}, \omega) = \int_{\partial \oplus} G_{in}(\boldsymbol{x}, \boldsymbol{\xi}, \omega) N_n(\boldsymbol{\xi}, \omega) \, \mathrm{d}\boldsymbol{\xi}^2,$$

(Aki and Richards, 2002), where summation over repeated indices is implied. The correlation of two such signals, averaged over an observation period, can be expressed by multiplication in the frequency domain, i.e.

$$\begin{aligned} C_{ij}(\boldsymbol{x}_1, \boldsymbol{x}_2, \omega) &= \langle u_i^*(\boldsymbol{x}_1, \omega) u_j(\boldsymbol{x}_2, \omega) \rangle \\ &= \left\langle \iint_{\delta \oplus} G_{in}^*(\boldsymbol{x}_1, \boldsymbol{\xi}_1, \omega) N_n^*(\boldsymbol{\xi}_1, \omega) G_{jm}(\boldsymbol{x}_2, \boldsymbol{\xi}_2, \omega) N_m(\boldsymbol{\xi}_2, \omega) \, \mathrm{d}\boldsymbol{\xi}_1 \mathrm{d}\boldsymbol{\xi}_2 \right\rangle, \end{aligned}$$

etc.

***Equation 3 has a minor typo.***
Thank you, the typo was corrected.

***Page 9, Line 240 is vague, and itself does not make much sense.***
The sentence in question stated: "The simulation in AxiSEM3D is run for a global mesh, but dropping the number of Fourier expansion coefficients to 0 at a depth of 800 km and at a distance above 90 degree." We understand that more detail is required to make the sentence less vague. We suggest to replace the sentence by the following short paragraph:

"AxiSEM3D represents the complex wavefield by Fourier expanding a 2-D model in the azimuthal direction. We simulated wave propagation with a high number of expansion coefficients in the region of interest, which allows the accurate representation of the model inside that region, while outside of it, the number of expansion coefficients was tapered to a minimum of two. This approach saves computational resources while also avoiding the use of absorbing boundary conditions. For details on the appropriate choice of the number of expansion coefficients, the interested reader is referred to Leng et al. (2019)."

Note that we used this particular approach because absorbing boundaries didn't ship with the published version of AxiSEM3D at the time the simulations were conducted.

***Figure 3 caption has autocorrelation, but all of the waveforms are from cross-correlations.***
This has been corrected.

**Section 5.2: Are the sources in Figure 6 simultaneously or randomly occurring. Is there any delay? And what is the frequency content?**
The sources are assumed to have converged to the expected value of a random, uncorrelated field of sources as in equation 3. Hence, in practice the source wavelets all have zero phase. We propose to extend the sentence in lines 325-326 as follows:
"The frequency content of the starting model is homogeneous for all sources (background and blobs), with Gaussian power spectral density S(xi, omega) of equation 3 having a mean frequency 0.05 Hz and standard deviation of 0.02 Hz."

We did not modify the figure caption itself, as it is already a little bit lengthy.

***Between 345 and 350 "wave forms" should be "waveforms".***
Thank you, this has been corrected.

***Figure 6: Rather than giving coordinates, please mark/number the selected stations and use them in the waveform plots.***
This is an excellent suggestion, and we have added a marker for the station pairs in question.

***I found the use of logarithmic signal ratio and asymmetry inversion confusing. Can you please describe it further or rephrase the part in Lines 310 onwards.***
Since Referee 1 also commented on this part, we have attempted to revise the paragraph describing the respective sensitivity kernels in lines 310 ff thoroughly, and we have included the equations describing both misfit functions (full waveform and asymmetry). We hope that this clarifies the usage of the asymmetry in the inversion. The paragraph now reads:

"Sensitivity kernels computed with noisi can be used to run gradient-based inversion for the distribution of ambient seismic sources from a data set of observed ambient noise cross-correlations. To demonstrate the effectiveness of this approach, we conduct two synthetic inversions using two different functions to measure the misfit between observations (here: synthetic data), and model. Firstly, we use the L2-norm of the synthetic (C syn ) and observed (C obs ) waveforms,

$$\chi_{\text{fwi}} = \frac{1}{2} \left[ \mathcal{C}^{syn} - \mathcal{C}^{obs} \right]^2 .$$

An exemplary sensitivity kernel for z-components of both receivers, and vertical sources,

$$K_{zz}(\boldsymbol{x}_1,\boldsymbol{x}_2,\boldsymbol{\xi}) = \int_{\omega=0}^{\omega_{Nyq}} G_{zz}^*(\boldsymbol{x}_1,\boldsymbol{\xi},\omega)G_{zz}(\boldsymbol{x}_2,\boldsymbol{\xi},\omega)f_{zz}(\boldsymbol{x}_1,\boldsymbol{x}_2,\omega)\,d\omega,$$

is shown in the left panel of Figure 5. It reveals how various locations of the source distribution affect the measurement. One can clearly recognize the pattern of stationary phase regions behind the stations and the oscillating sensitivity in between the stations (e.g. Snieder, 2004; Xu et al., 2019). In contrast, the right panel of Figure 5 shows sensitivity Kzz of another misfit function,

$$\chi_A = \frac{1}{2}\left[A(\mathcal{C}^{syn}) - A(\mathcal{C}^{obs})\right]^2,$$

where

$$A(x(\tau)) = \ln\left(\frac{\int[w_+(\tau)x(\tau)]^2 d\tau}{\int[w_-(\tau)x(\tau)]^2 d\tau}\right),$$

(Ermert et al., 2016), and w + , w − denote causal and a-causal window, respectively. For simplicity, we will refer to this as asymmetry in the following. This second sensitivity kernel is smoother than the full-waveform one: The oscillating sensitivity between the stations is removed due to the averaging over w− , w+ , and the stationary phase regions have opposite signs of sensitivity due to the ratio:"

$$\frac{\int[w_+(\tau)C(\tau)]^2 d\tau}{\int[w_-(\tau)C(\tau)]^2 d\tau}.$$